# High-Entropy Coatings (HEC) for High-Temperature Applications: Materials, Processing, and Properties

**Muhammad Arshad, Mohamed Amer** , **Qamar Hayat** , **Vit Janik, Xiang Zhang** , **Mahmoud Moradi** and **Mingwen Bai** *

Institute for Clean Growth and Future Mobility, Coventry University, Coventry CV1 5FB, UK;
arshadm15@uni.coventry.ac.uk (M.A.); amerm5@uni.coventry.ac.uk (M.A.); hayatq@uni.coventry.ac.uk (Q.H.);
ac6600@coventry.ac.uk (V.J.); xiang.zhang@coventry.ac.uk (X.Z.); ad6683@coventry.ac.uk (M.M.)
* Correspondence: mingwen.bai@coventry.ac.uk

**Abstract:** High-entropy materials (HEM), including alloys, ceramics, and composites, are a novel class of materials that have gained enormous attention over the past two decades. These multi-component novel materials with unique structures always have exceptionally good mechanical properties and phase stability at all temperatures. Of particular interest for high-temperature applications, e.g., in the aerospace and nuclear sectors, is the new concept of high-entropy coatings (HEC) on low-cost metallic substrates, which has just emerged during the last few years. This exciting new virgin field awaits exploration by materials scientists and surface engineers who are often equipped with high-performance computational modelling tools, high-throughput coating deposition technologies and advanced materials testing/characterisation methods, all of which have greatly shortened the development cycle of a new coating from years to months/days. This review article reflects on research progress in the development and application of HEC focusing on high-temperature applications in the context of materials/composition type, coating process selection and desired functional properties. The importance of alloying addition is highlighted, resulting in suppressing oxidation as well as improving corrosion and diffusion resistance in a variety of coating types deposited via common deposition processes. This review provides an overview of this hot topic, highlighting the research challenges, identifying gaps, and suggesting future research activity for high temperature applications.

**Keywords:** high-entropy coatings; high-temperature applications; functional properties; oxidation; corrosion; thermal barrier coatings

## 1. Introduction

A thermodynamic system becomes energy efficient when it can be operated at high temperature. In the power generation industry, any increase in operating temperature reduces fuel consumption and carbon emission similarly; in the jet engine industry, an increase in the operating temperature delivers heavier payloads, greater speed, and greater range [1]. Therefore, it becomes essential to develop materials to perform well in harsher environments. There is a general acceptance that the performance of any alloy system is limited by the base metal/element properties [1]. As the system becomes more heavily alloyed, it results in a higher chance of the formation of deleterious phases (i.e., intermetallic or Lavas) [2]. An innovative approach to design a thermodynamically stable multi-element system was first reported in 2004 by Yeh et al. [3] and Cantor et al. [4] with the introduction of a new concept of High-Entropy Alloys (HEAs). The HEAs formed reported no formation of intermetallic, Laves or other topologically close-packed (TCP) phases. This gave rise to significant interest in HEMs and stimulated enormous research and development in many applications [5,6]. This novel design strategy is conducted by mixing at least five

elements/components [7] in equi-atomic or unequi-atomic ratio (5–35% variation in elements concentration) based on the desired application. Until today, approximately more than 300 HEAs have been processed with more than 35 principal components [8].

The uniqueness of HEMs comes from its complex composition resulting in several phenomena, which can be explained by four core effects: (1) Thermodynamics: high-entropy effects; (2) Kinetics: sluggish diffusion; (3) Structures: severe lattice distortion; and (4) Properties: cocktail effects [9]. These phenomena in HEAs render superior mechanical and functional properties [10], including high strength, thermal stability, hardness and ductility, oxidation, and corrosion resistance [11–14]. All of these have made HEMs one of the most competitive candidate materials that have the potential to meet the essential requirements of materials to service in the complex, harsh, and highly sensitive service environments, such as nuclear fuels rods, turbine blades and pressure vessels [15]. The wide compositional range and great flexibility in elements selection of high-entropy materials (HEM in a more general term) have now opened a new area for materials scientists and engineers to explore their value both as structural and coating materials [16].

In the past decade, HEMs have been widely applied in every important sector as a coating material and is still widely researched for practical application. The research progress in the field of HEC has been summarised in several review papers, discussing fabrication, processing, characterisation, and properties assessment. This article provides a succinct review of the research progress of HEC particularly for high-temperature application. A thorough review of the materials classification, coating fabrication processes, its complex relationship among microstructure, phases and properties are discussed. It is not the intention of the authors to give a complete bibliography of the processing and properties of HEC but rather to point out the main trends and open questions in this field, which are frequently illustrated by results from publications that clearly demonstrate the suitability of processes and materials for high-temperature use.

## 2. HEC Materials Classification

Currently, we have numerous high entropy-based materials formed through the combination of various elements and investigated for their properties based on application. Here, high entropy materials have been classified in a simplified pictorial representation in Figure 1, which is based on the current explored composition space and the degree to which it has been researched [17,18]. The classification of HEMs is presented here as a grouping of elements with common characteristics or that to produce a particular group of properties. Many HEC works are focused on metallic HEA, which can be divided into two further groups mainly based on their elemental positions in the periodic table. A wide range of different HEMs used for HECs from the literature are summarised in Table 1.

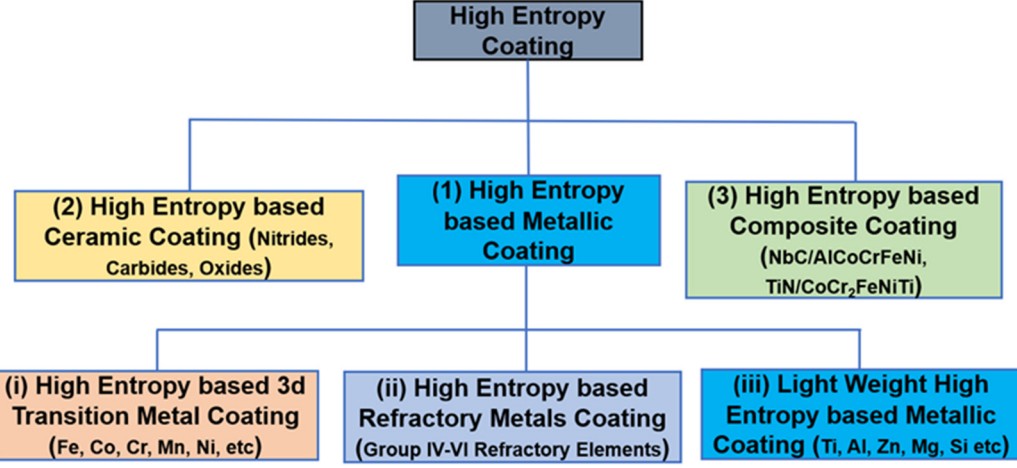

**Figure 1.** Classification of high-entropy coatings based on composition (3d stands for 3d block of elements, while groups IV–VI are refractory elements in the periodic table).

**Table 1.** Summary of different high-entropy-based coating materials deposited by conventional coating deposition techniques investigated for desired properties.

| HEC Type | | HEC Composition | Substrate | Deposition Process | Phase/Structure | Temp (°C) | Outstanding Characteristics/Properties Exhibited | Ref |
|---|---|---|---|---|---|---|---|---|
| High Entropy Metallic coatings | 3d Transition Metal Coatings | NiCoCrAlTi | Haynes 230 | HVOF | FCC, BCC | 1150 | Excellent oxidation resistance, | [19] |
| | | NiCo$_{0.6}$Fe$_{0.2}$Cr$_{1.5}$SiAlTi$_{0.2}$ | 304-SS/Inconel 718 | Plasma Spraying/ HVOF/SPS | BCC, FCC, Cr$_3$Si-phase | 1100 | High oxidation resistance, High wear resistance, High hardness, Lower thermal conductivity, | [20–23] |
| | | CuAl$_x$NiCrFe | Inconel 718 | Laser Cladding | FCC, BCC | 1100 | High thermal stability, High oxidation resistance, High diffusion resistance | [24] |
| | | NiCrCoTiVAl | Ti-6Al-4V | Laser Alloying | BCC, Intermetallic | ≤1005 | High oxidation resistance, High thermal stability | [25] |
| | | Al$_x$CoCrFeMnNi | Q235 Steel | Plasma Cladding | FCC, BCC | 600 | High corrosion resistance, High oxidation resistance, High wear resistance | [26] |
| | | AlTiCrNiTa | Zr-4 | Magnetron sputtering | FCC, amorphous | 330 | High hardness, High corrosion resistance | [27] |
| | | Y-Hf doped AlCoCrFeNi | Ni-super alloy | Sintering process | FCC, BCC | 1100 | Low oxidation rate, High oxidation resistance | [28] |
| | | AlCoCrFeNi | Ni-super alloy | Cold Spray | FCC, BCC | 1100 | High thermal stability, High oxidation resistance, | [29] |
| | | (CoCrFeMnNi)$_{0.85}$Ti$_{0.15}$ | Q235 Steel | Plasma Cladding | FCC, BCC, Intermetallic (sigma) | 400 | High hardness, High wear resistance | [30] |
| | Refractory High Entropy Metal Coatings | MoFeCrTiWAlNb | M2-Steel | Laser Cladding | BCC, HCP | 600 | High Hardness, High wear resistance | [31] |
| | | AlTiVMoNb | Ti–6Al–4V | Laser Cladding | BCC | 800 | High Hardness Oxidation Resistance > Substrate Light weight | [32] |
| | | TiZrNbWMo | 45-Steel | Laser cladding | BCC, TiW ppt | 800 | High hardness, High thermal stability | [33] |
| High Entropy Ceramic Coatings | | RE$_2$(Ce$_{0.2}$Zr$_{0.2}$Hf$_{0.2}$Sn$_{0.2}$Ti$_{0.2}$)$_2$O$_7$ (RE = Y, Ho, Er, Yb) | Ni-super alloy | APS | Fluorite | 1200 | High thermal expansion coefficient, Low thermal conductivity, High hardness | [34] |
| | | (La$_{0.2}$Nd$_{0.2}$Sm$_{0.2}$Eu$_{0.2}$Gd$_{0.2}$)$_2$Zr$_2$O$_7$ | Ni-super alloy | APS | Fluorite | 1100 | High thermal stability, High coefficient of thermal expansion. | [35] |
| | | AlCoCrCuFeNi/Mg-alloy | AZ91D | Laser cladding | a-Mg, intermetallic, BCC | - | High wear resistance | [36] |
| | | (AlCrMoTaTi)N | p-Si (100) | Magnetron sputtering | FCC, BCC, amorphous | 800 | High electrical resistivity | [37] |
| High Entropy Composite Coatings | | CoCr$_2$FeNiTi$_x$/TiN | 904L-steel | Laser cladding | FCC, TiN, Laves phase | - | High wear resistance, Low corrosion resistance | [38] |
| | | AlCoCrFeNiTi/Ni60 | 316-SS | Plasma Spraying | FCC, BCC | 500 | High wear resistance | [39] |
| | | AlCoCrFeNi/NbC | Q235-Steel | Laser cladding | FCC, BCC | - | High hardness, High wear resistance | [40] |

- HEM: high-entropy materials;
- HEA: high-entropy alloys;
- RHEA: refractory high-entropy alloys;
- HEC: high-entropy coatings;
- HEAC: high-entropy alloy coatings;
- RHEAC: refractory high-entropy alloy coatings;
- HECeC: high-entropy ceramic coatings;
- HECoC: high-entropy composite coatings;
- LWHEC: light weight high-entropy coatings.

### 2.1. HEA Metallic Coatings

#### 2.1.1. 3d-Transition Metal

The earliest research on HEAs was based on the 3d-transition metal alloys with an element selection of Fe, Co, Cr, Mn, and Ni, also known as Cantor alloys [4] (see Figure 2). To date, the most studied HEA family for high-temperature coating applications contains elements of Cr, Mn, Fe, Co, Ni, Cu, Al, Ti, W, and V, of which 85% of the alloys contained four elements of the 3d-transition metals [7,41–46]. These base elements retain the phase composition while functional elements such as Al, W, Ti, and V are applied to further improve their thermomechanical and functional properties. The HEAs systems comprising "Cr–Fe–Co–Ni" elements have been developed into a wide range of HEA systems. These alloys were found to have enhanced mechanical properties at up to 800 °C [47].

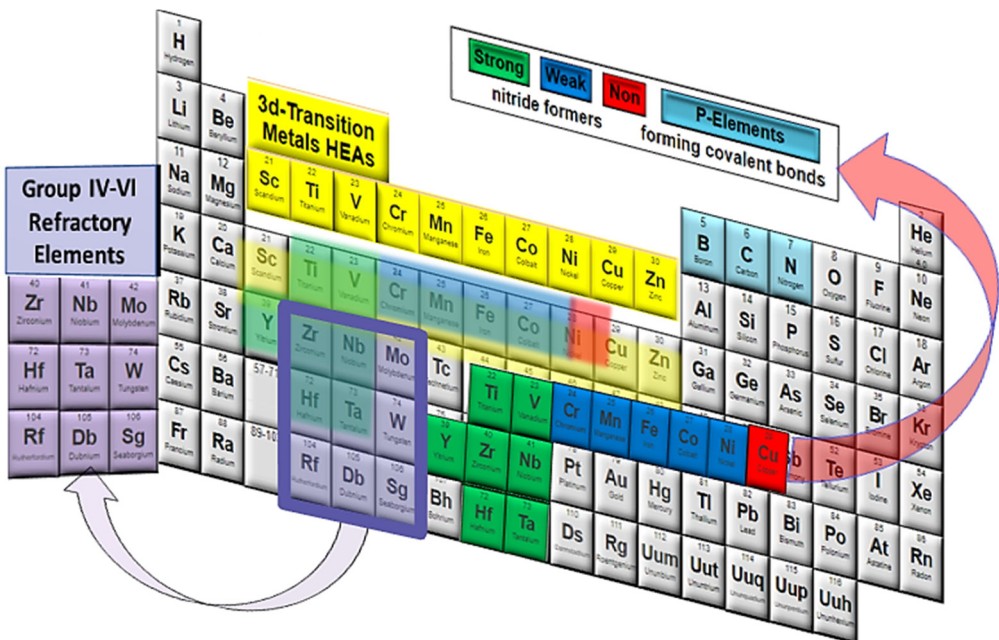

**Figure 2.** Periodic table of elements with highlighted elements that are most frequently used in the design of 3d-transition metal HEAs, refractory high-entropy alloys (RHEAs) and high-entropy ceramics (HECe). Reprinted with permission from [17] Copyright 2017 Elsevier.

In a liberal view, HEAs based on 3d-transition metals can be regarded as an extension of Ni superalloys and stainless steels. Cr-Fe-Ni elements are also the principal elements in austenitic (FCC), duplex (FCC + BCC) and precipitation hardened stainless steels [17,47]. In some austenitic steels, Mn is also a principal element with the addition of Al, Cu, Nb and Ti to precipitation-hardened steels. In addition, Ni-superalloys are having complex compositions, which are based primarily on Ni-Cr-Fe-Co-Mo with Al, Ti, or Nb, to form a considerable volume of intermetallic phases of composition $Ni_3(Al, Nb, Ti)$ [17].

In addition, HEAs should generally have low thermal conductivities due to their complicated distribution of solute atoms, which should be beneficial for the high-temperature applications. Hsu et al. [20–23] demonstrated the low thermal conductivity of HEAC $NiCo_{0.6}Fe_{0.2}Cr_{1.5}SiAlTi_{0.2}$ as a heat-resistant coating for high-temperature application as an alternative to MCrAlY, which is used as a bond coat in the TBC system. In cyclic oxidation test at up to 336 cycles and 1100 °C HEAC, it showed good high temperature oxidation resistance due to the formation of a continuous and adherent $\alpha$-$Al_2O_3$ layer on the coating surface. The authors reported 30% lower thermal conductivity of HEAC in comparison to MCrAlY at high temperature, which is beneficial for improving the heat insulation of the TBC system. Another important parameter, the thermal expansion coefficient of HEC, was found to be lower than MCrAlY, which makes it more compatible as a bond coat in avoiding interface stresses.

### 2.1.2. Refractory High-Entropy Alloy (RHEAs)/Group IV–VI Metal Alloys

RHEAs are mainly composed of group IV–VI refractory elements of the periodic table (highlighted purple in Figure 2) that mostly form BCC structures [9]. Some of the main alloying elements of RHEAs are Hf, Mo, Nb, Ta, Rf, and W of much higher melting temperatures (Tm) and Al, Si non-refractory elements are added for density reduction and properties improvement [38]. At the US Air Force Laboratory were the first who reported quaternary MoNbTaW and quinary MoNbTaWV RHEAs with high compressive strengths up to 1600 °C compared with conventional superalloys [39,40].

The development of RHEAs has been stimulated by two main factors: (1) to overcome the limitation of Cantor alloys and (2) to surpass the properties offered by superalloys [2]. The 3d-transition HEAs failed to retain strength at temperatures above 800 °C [48]. To retain mechanical properties and demonstrate oxidation resistance, a small number of RHEAs have been investigated: $Al_{0.3}Ta_{0.8}NbTi_{0.4}V_{0.2}Zr_{0.3}$ [49], $Al_{0.5}Ta_{0.8}NbTi_{0.5}V_{0.2}Zr$ [50], $Al_{0.3}TaNbTi_{0.4}Zr_{0.3}$ [51], and $CrNbMo_{0.5}Ta_{0.5}TiZr$ [52], all of which have demonstrated great potential towards these goals.

Similarly, RHEAs as coating materials are designed to resist substrates high-temperature oxidation, corrosion, and abrasion. The RHEAs compositions that are most often used as coatings are MoNbTaW, HfNbTaZr, CrMoNbTa, NbMoTaWV and CrNbZrV, having a single-phase BCC structure and hold high strength (>400 MPa) at temperatures up to 1600 °C [47]. However, RHEA oxidation resistance is not sufficient because R elements (i.e., Ti, Zr and Hf) shows strong oxygen affinity results in the formation of non-protective oxide layers, while oxides of V, Mo and W have low melting/boiling points [53].

### 2.1.3. Light Weight High-Entropy Metals

The importance of lightweight materials with outstanding mechanical properties has made researchers explore alloys based on elements such as Ti alloys, Mg alloys and Al alloys [54]. As these alloys are based on one or two elements, that limits further exploration. In this regard, the concept of high entropy opens up new possibilities for new lightweight alloys development. The lightweight high-entropy alloys in group I were developed by rationally selecting light weight elements that include Al, Be, Mg, Li, Ti, Zn, etc. [55]. These LWHEAs have ultra-low specific density (1–1.6 g/cm$^3$): for instance, $Mg_x(AlMnZnCu)_{100-x}$ LWHEA with a density of (2.2–4.30 g/cm$^3$) [56] and AlLiMgSiCa LWHEA with a density of 1.4–1.7 g/cm$^3$ [57]. These LWHEAs normally contain various brittle precipitates such as ordered phase and intermetallic compounds, which limits its mechanical properties. The LWHEAs in group II are designed from the existing solid-solution HEAs. The pre-existing solid solution phases are inherited in LWHEAs, for instance, replacing Hf and Ti (heavy elements) in TiZrHfNbTa HEA with V and Al (light weight elements) a series of $Ti_aZr_bV_cNb_dAl_e$ LWHEAs with body-centred cubic phase were designed [58,59]. Based on the same strategy, two series of LWHEAs with single/main BCC phase, $Ti_x(AlCrNb)_{100-x}$ and $TiZrV_{0.5}Nb_{0.5}$, were designed, both of which show considerable tensile strength–ductility matching [60].

### 2.2. High-Entropy Ceramic Coatings

In recent years, high-entropy ceramics (HECe) have been gaining attention since Rost et al. [61] adapted the idea of HEA to design the first entropy-stabilized oxide [62]. In HECe, the transition metal of HEAs combines with C, N, B, and other negatively charged elements of p-block (highlighted in Figure 2) [63] forming highly stable nitrides, oxides, carbides, and carbonitrides (due to ionic and covalent bonding) occupying voids in a closed packed structure [64]. In the periodic table, elements of group 3–5 form strong nitrides (highlighted green in Figure 2), while transition metals in groups 7–11 are either weakly nitride formers (blue in Figure 2) or non-nitride formers (red in Figure 2) [65].

High-entropy oxides (HEOs) are known for their low thermal conductivity, which is caused by phonon scattering by multi-components and distorted lattices [66]. In $RE_2TM_2O_7$ high-entropy oxides, RE stands for the rare-earth elements, such as La, Ce, Y, Yb, etc., and

TM stands for transition metals that can be used as environmental/thermal barrier coatings (EBC/TBC) due to their low thermal conductivity, high thermal stability, and their tunable thermal expansion coefficients [67–72]. It was predicted that the thermal conductivity decreases by increasing the size disorder factor due to the significant effect of severe lattice distortion on reducing the thermal conductivity [73]. The beneficial effect caused by HE has also been applied to other ceramics including silicide, boride, nitride, and carbide, which all demonstrated lower thermal diffusivity and conductivity [74–77].

High-entropy ceramic coatings (HECeC) are deposited through reactive coating deposition methods such as magnetron co-sputtering in which target atoms/ions react with the atmosphere of $N_2$, $CH_4$, $O_2$, and $CH_4 + N_2$ [37,78–80]. The transition metal bulk nitrides formed normally have high hardness, resistance to wear, and refractory character with melting temperature exceeding 1800 °C [81] with some over 4000 °C [82]. Similarly, deposited HECeCs are described to have superior surface properties, which include thermal stability, high hardness, anti-corrosion, and diffusion resistance [83,84], which have potentials as diffusion barriers as well as hard protective coatings [85]. The mixed bonding character of transition metal nitrides, carbides, and carbonitrides which is partially covalent, ionic, and metallic gives rise to maintaining the high melting temperatures and hence thermal stability. The high diffusion resistance of ceramic coatings is predominantly attributed to the amorphous structure with nano-crystalline, the multiple-element effect, and the high stacking density with no rapid diffusion path [86]. It proves that the HECeCs based on appropriate compositions and processing parameters surpass conventional binary coatings of nitride, carbide, and carbonitride, primarily owing to HEMs' four core effects [1,87].

### 2.3. High-Entropy Composite Coating

High-entropy composite coatings (HECoC) are attractive as advanced coatings used for addressing specific requirements by utilizing the combined effect of two or more constituents having the ability to form layered or mixed structures. The production of HECoC further broadens HEM's application as coating materials; HEAs acting as a binder or matrix reinforced with hard ceramics, or another way round, it acts as reinforcement in low-density alloys (i.e., Mg and Al) [88]. TiN [38], NbC [40], TiC, and TiB2 [89] hard ceramic reinforcement with high melting temperature and hardness, chemical stability, and excellent wear resistance, forming a strong metallurgical bond with the HEAs matrix, have been produced as composite coatings. Some progress has been accomplished in HECoCs: for instance, the $CoCr_2FeNiTi$ /TiN [38], AlCrCoFeNi/NbC [40] coatings produced via laser cladding and used in situ plasma transferred arc cladding for $TiC–TiB_2$/CrCoCuNiFe coatings [89]. Ni60 powders (i.e., hard nickel-based alloys with excellent wear resistance, hardness, and corrosion resistance) combined with hard ceramic reinforcements were added to AlCrCoFeTiNi HEA, deposited by plasma spraying [39] and resulted in a coating that exhibited high hardness and anti-wear properties at high temperature.

A composite coating composed of Mg/Al matrix reinforced with HEA particulate could take full advantage of HEA particles' characteristic properties (i.e., hardness, anti-wear, and thermal stability [39]), the lightweight Al/Mg matrix [90] and the matrix/HEA's great physicochemical compatibility. Accordingly, the laser melt injection technique used for AlCrCoCuFeNi particles-reinforced Mg matrix composite coating [36,91] had a noticeable improvement in tribological properties. Mechanical alloying was used for the preparation of the Al–CrCoMnFeMoW composite coating, in which the formation of oxide film improved the high-temperature oxidation resistance of Ti substrate [92].

In summary, the marked outstanding properties of HEMs make them a potential candidate for high-temperature applications. This article reviewed the advancement in the field of HEC regarding its potential as high-temperature coating materials. HEAs metallic coatings (3d-transition metals and refractory metals), HECe coatings (ionic and covalent) and HECo coating (HEA matrix and HEA reinforcement) are the three basic HEC material types in the available literature. Articles on HEA metallic in the form of bulk as well as coatings are mainly focused on producing single-phase HEA with few attempts to produce

multi-phase coatings. On the contrary, the current high-temperature conventional materials such as Ni-superalloy achieve its strength and high temperature properties due to the presence of the second phase. There is scope for exploring HEM coatings with a dual-phase microstructure or composite microstructure or the dispersion of stable oxides to achieve enhanced high-temperature properties.

According to the current research progress, the classification of HEMs is limited to the five different HEMs, which are grouped on their distinctive properties. Future research and findings will further improve classification systems and will be updated or revised with HEMs research progress and development in surface modification methods.

## 3. HEC Fabrication Methods

Most of the coating fabrication processes fundamentally rely on one of the three general coating deposition processes: chemical vapour deposition, physical vapour deposition and thermal spray processes. The features, advantages/disadvantages of the coating methods most relevant to high-temperature HEA coatings are summarised.

### 3.1. Laser/Plasma Direct Deposition Methods

A strong metallurgical bond between substrate and coatings is achieved with laser deposition processes including laser surface alloying, laser cladding, etc. These processes have the advantage of high energy density, producing coatings of fine and uniform microstructure besides strong metallurgical bonding (i.e., coatings and substrate) [93,94]. In these processes, the ultra-high heating/cooling rates cause minute damage to the substrate [95] during deposition and are currently widely used for HEA coatings. Figure 3 shows the schematic drawing of the laser deposition process in which a laser irradiates substrate with the evenly distributed preplaced powders in the protective environment [96].

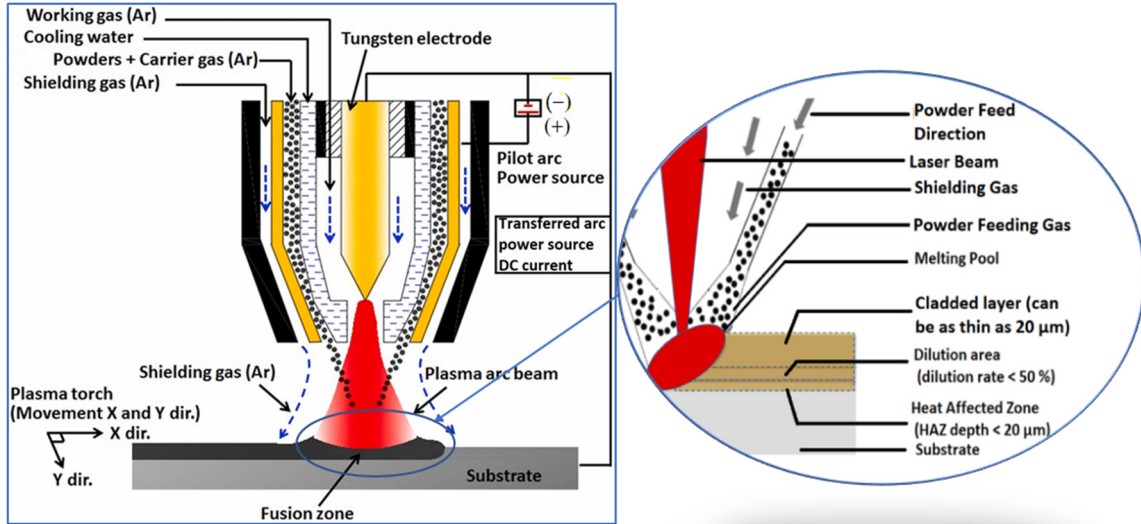

**Figure 3.** Shows laser surface alloying with a magnified image of the processes during laser surface alloying. Reprinted with permission from [97,98] Copyright 2020 Elsevier.

### 3.1.1. Laser Cladding

Strong metallurgical bonding between the substrate and cladded coatings is achieved through coating/substrate dilution (>5%). High-entropy metallic and composite coatings are laser-cladded on a variety of substrate materials such as Ti–6Al–4V [32], Ni-superalloys [24], steels [31,99], etc., having a thickness of hundreds of μm to several mm. Despite the use of different kinds of substrates materials, research is focused on the characterisation, microstructure evaluation, and surface properties, based on HEAs design and processing optimisation, their effects on phase constitution, and desired properties. The microstructure

of laser-cladded HEA deposited coatings is fine and homogeneous with amorphous and nanocrystalline structure formation due to its ultra-fast heating and cooling rate [14].

Properties of laser cladding coating can be tailored with the optimisation of processing parameters (i.e., laser scan speed, laser power, material feeding rate, laser spot size, and hatch distance), among which the laser power has a vital influence on the microstructure and resulting mechanical/functional properties of deposited coatings [100–104]. CuAlNi-CrFe HEA coating was deposited as a bond coat on Inconel 718 superalloy with ultra-speed laser cladding [24]. Apart from a strong metallurgical bond between the HEA bond coat and substrate, a short pre-oxidation time is required as suggested which can reduce initial oxidation and avoid the formation of non-protective oxides (i.e., spinel) as shown in Figure 4. A block-like structure instead of a lamellar structure in thermal spraying ensured continuous and sufficient supply of Al and improved oxidation resistance at high-temperatures. Of HEAs' four core effects, the sluggish diffusion lowered TGO growth rate and minimized substrate/coating inter-diffusion [97].

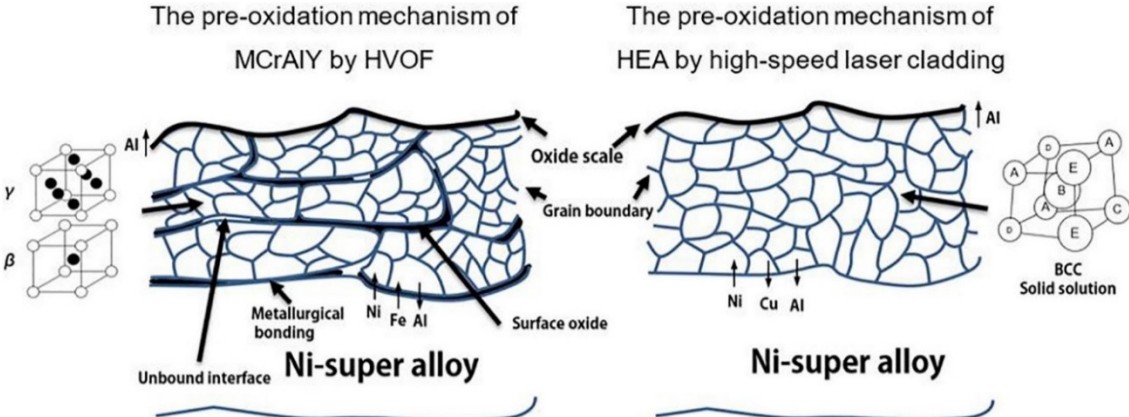

**Figure 4.** Shows difference of pre-oxidation between HVOF-deposited conventional MCrAlY and high-speed laser -cladded HEA coating. The blocking effect to the continuous depletion of Al element in lamellar structure in the thermal-sprayed coating is due to the unbound interface, surface oxide and partial metallurgical bonding. Reprinted with permission from [97] Copyright 2020 Elsevier.

The increasing demand for high-speed cutting in dry conditions of special alloys has resulted in higher red-hardness for the tools (i.e., the maximum temperature at which steel retains a given hardness). Martensite and carbide precipitates improved the steel tools' hardness and wear resistance above 600 °C [105,106]. MoFe$_x$CrTiWAlNb$_y$ RHEA was investigated as a potential tool coating material for special machining processes [107]. The RHEAs coatings were prepared by laser cladding on an M2 substrate. Fe$_x$Nb$_y$ (x = 1, 1.5, y = 1) HEA was reported as an optimal alloy coating for laser cladding with minimum defects and a strong metallurgical bond with the substrate. The as-deposited coating was mainly composed of body-centred-cubic solid solution MC carbides and unmelted W-particles. The defect-free Fe$_{1.5}$Nb$_1$ showed a high hardness of 913 HV during high-temperature annealing (800 °C), while the dendrites and MC carbides remained stable. In addition, the deposited coating exhibited superior oxidation resistance at 800 °C in comparison to M2 steel demonstrating great potential in real industrial applications.

### 3.1.2. Laser Surface Alloying

The degree of coating dilution makes a distinction between laser surface alloying and laser cladding. As the cooling rate in these processes is usually high (i.e., $10^4$–$10^6$ K/s) [108], it restricts the long-range diffusion process, suppressing the grain growth during solidification, resulting in a non-equilibrium phase formation and fine microstructure. HEA metallic coatings with Cr, Co and Ni as principal elements deposited as coatings on a range of substrates, including steel [109], Cu [110], Ti [111,112], Al [113], and Ni super-alloy [114].

The application of laser surface alloying for HEA coating deposition for high-temperature application is limited by the inevitable dilution during processing. In a study, laser surface alloying is used for the deposition of NiCrCoTiVAl HEA on Ti6Al4V substrate [114]. Encouraging results were obtained for phases' thermal stability when the coating was exposed to a temperature of 900 °C for 8 h, while the constituted phase of BCC and compound (Ni, Co) $Ti_2$ remained unchanged. However, owing to the continuous dilution of the Ti element, the BCC lattice parameter of the phase increased from 3.06 to 3.16 Å. The oxidation film on the NiCrCoTiVAl HEA coating is mainly composed of $TiO_2$, $V_2O_5$ and NiO, which is responsible for the mass increase. The synthesized HEA coating opposed to the substrate was proved to be stable below 1005 °C and was confirmed by differential scanning calorimetry and the dynamic differential scanning calorimetry [25].

### 3.1.3. Plasma Cladding

The characteristic features of plasma cladding include very high energy density, minimum thermal deformation, low substrate dilution, and comparatively low cost, which has attracted attention for its application in the field of HEA coating [115–119]. Contrary to laser surface alloying and laser cladding, a higher heat input and higher blowing force of plasma cladding allows for maximum melting and the mixing of coating materials, resulting in a homogenized microstructure and excellent performance [120]. The highlights in the available literature are mainly on the coating design for improved surface performances.

Plasma cladding was used for the deposition of CoCrFeMnNi and $(CoCrFeMnNi)_{85}Ti_{15}$ HEA coating, and researchers investigated its microstructure, mechanical properties, and tribological performance [121]. The phase analysis of CoCrFeMnNi HEA coating contained an FCC solid solution, while the $(CoCrFeMnNi)_{85}Ti_{15}$ HEA coating included both FCC and BCC solid solutions with some quantity of an intermetallic sigma phase. Figure 5 shows a comparison of wear rates of the two coating surfaces. The wear experiment was completed from room temperature (RT) to 800 °C against $Si_3N_4$ balls at 25 N load conditions. The exceptional hardness of the $(CoCrFeMnNi)_{85}Ti_{15}$ HEA coating was attributed to the hard BCC solid solution and sigma phase with hardness six times (i.e., $150.1 \pm 7.4$, and $910.5 \pm 26.6$ HV) higher than that of the CoCrFeMnNi HEA coating. Furthermore, the tribological property–wear rate of the $(CoCrFeMnNi)_{85}Ti_{15}$ HEA coating was reported to be nearly six times that of the CoCrFeMnNi HEA coating during the high-temperature friction process. The wear resistance was observed to increase for $(CoCrFeMnNi)_{85}Ti_{15}$ alloy up to 400 °C working temperature. To 800 °C, oxidation and contact fatigue were the dominant wear mechanisms for the $(CrCoFeMnNi)_{0.85}Ti_{0.15}$ alloy. Oxidation and adhesive wear were the main wear mechanisms above 800 °C and hence, the wear resistance is reduced significantly at 800 °C.

Plasma cladding [26] was applied to study the effect of Al content on the microstructure, wear, corrosion, and high-temperature oxidation resistance of CoCrFeMnNi HEA coating. It is reported that the $Al_xCoCrFeMnNi$ HEA coating cladded had a dendritic structure and retained the FCC phase when the Al content in alloy x ≤ 0.5; upon x ≥ 1.0, the coating had two phases of FCC and BCC. Hardness and wear resistance was reported to increase with Al increase in $Al_xCoCrFeMnNi$ alloy coating. Metal ions diffusion controlled the entire process of high-temperature oxidation. At 900 °C high-temperature oxidation, Mn outward diffusion in the coating layer controlled the entire process, and the oxide-film key component was a Mn-rich non-densified oxide. In addition, a finer and dense oxide $(Al_2O_3/Cr_2O_3)$ film well below the oxide film $((Al_2O_3, Mn_3O_4, Cr_2O_3,$ and $(Ni, Mn)(Fe, Cr, Al)_2O_4))$ produced with an increased Al content. The oxide film reduced the diffusion of base elements and oxygen and hence, the oxidation resistance is improved at high temperatures.

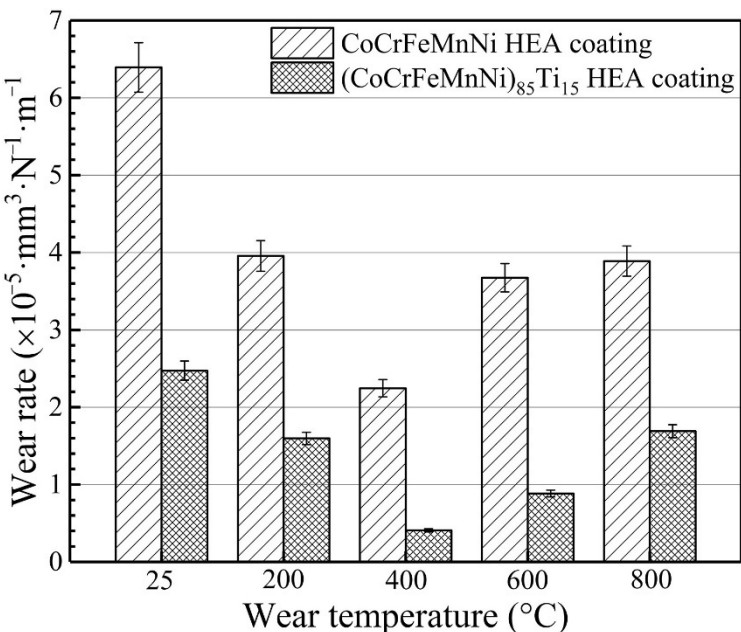

**Figure 5.** Two HEAs coating types of wear rate at different wear temperatures. Reprinted with permission from [30] Copyright 2020 Elsevier.

In summary, the complex process–structure–property relationship remains the focus of research in surface coating/modification processes. In HECs research for high-temperature applications, both microstructure and mechanical/functional properties are investigated, which are mainly dependent on the coating composition and optimization of the process parameters. In this regard, laser cladding, laser surface alloying and its variants have great potential for HECs deposition due to its advantage of higher energy density. However, the most common problems of cracks and pores are the inherent characteristics of these techniques due to high residual stress and shielding gas caused by the high heating and cooling rates. Furthermore, the raster scan of the laser causes individual track overlap that leads to the deposited layer heat treatment with change in its microstructure and is sometimes unfavourable to the coating performance.

*3.2. Thermal-Spraying Processes*

As invented a century ago, thermal spray (TS) has been continuously developed based on inherited flexibility, which has led to the expansion of its variants for coating surfaces and applications in a variety of fields. In TS processes, coating precursors, in different physical states (i.e., molten, or semi-molten), are accelerated onto the surface of the substrate producing dense/thick coatings [122–124]. The schematic shown in Figure 6 represents a general thermal spraying process and coating produced. Flexible thermal sources, a variety of feedstocks, and jet configurations resulted in different types of spraying, including arc, flame, HVOF (high-velocity oxygen-fuel), plasma, and cold spraying, which are used to protect the substrate against wear, oxidation, corrosion, and heat in a high-temperature environment [122,125,126]. Plasma, HVOF, and cold spraying have recently been employed to fabricate HECs with improved mechanical and functional performances. The chemical composition of the feedstock is not the only main factor of the thermally sprayed HEMs that determine the coating performance; spraying process parameters also play a major role. Therefore, the surface performance of HECs corresponding to different microstructures produced by plasma, HVOF, and cold-spraying methods for high-temperature applications are reviewed in this section.

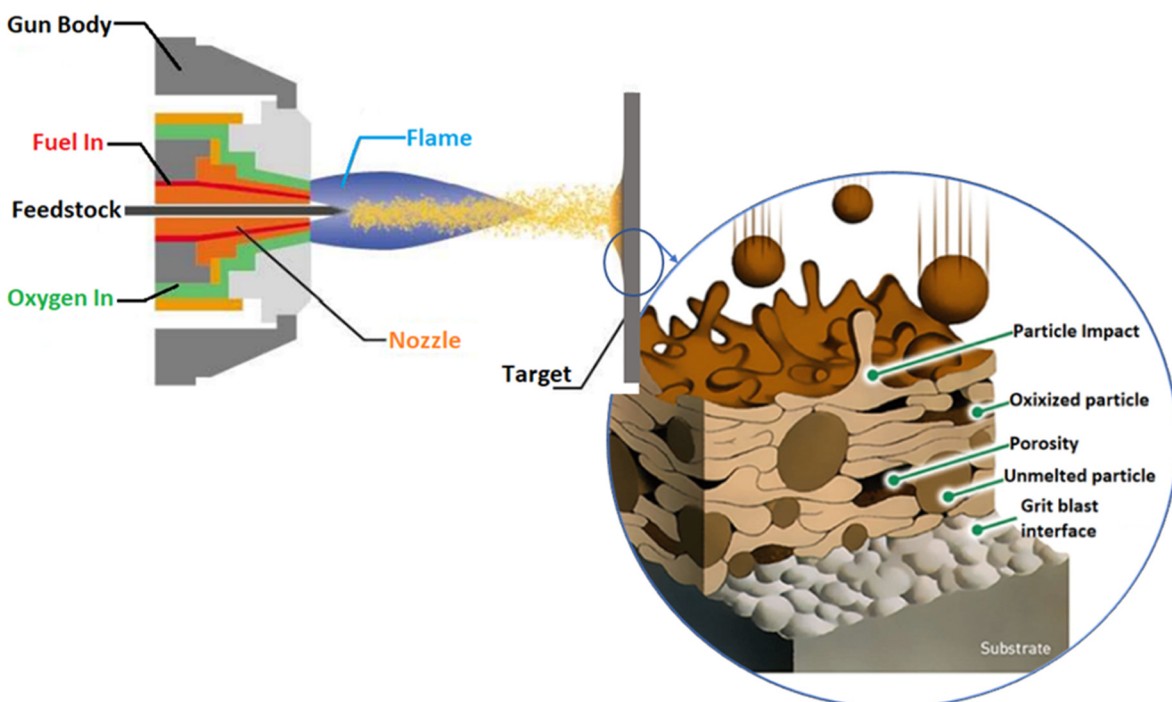

**Figure 6.** Illustration of a thermal spray process with a magnified image highlighting various features in the coating. Reprinted with permission from [127] Copyright 2020 Elsevier.

### 3.2.1. Plasma Spraying

The plasma spraying (PS) process is reported to be extensively applied in aerospace, automotive, petrochemicals, and mining industries for coating production and repair workpieces [128–130]. In this process, high-temperature plasma is generated by the gas (i.e., Ar) passing through a high-voltage arc causing gas molecules to split into ions and electrons. The high heat of PS deposits a variety of coating materials, including high melting point materials such as ceramics. Different types of PS variants are based on the plasma environment such as vacuum plasma spraying (VPS), atmospheric plasma spraying (APS) and low-pressure plasma spraying (LPPS). For HECs fabrication, laser cladding and PVD are widely used techniques, but recently, PS has been reported to fabricate HECs [39,131] because of its inherent advantages of concentrated high flame temperatures, strong coating adhesion, high deposition efficiency and minimum substrate/coating dilution.

High-temperature mechanical and oxidation resistance properties were investigated for $NiCo_{0.6}Fe_{0.2}CrSiAlTi_{0.2}$ HEA [20], which is an overlay coating on 304 stainless steel substrate deposited by APS. The coatings showed great potential as a suitable substitute for MCrAlY overlay coating, which is usually favoured for structural components protection at high temperature but has poor wear performance in conditions involving hard particulates in an operating environment. The coatings presented a lamellar structure of supersaturated BCC phase and showed oxidation resistance as good as a typical NiCrAlY coating at 1100 °C with the formation of dense TGO comprising of $Al_2O_3$ and $Cr_2O_3$. $NiCo_{0.6}Fe_{0.2}CrSiAlTi_{0.2}$ coating hardness surpassed MCrAlY coating when aged at 800 °C due to $Cr_3Si$ precipitate formation that increased its hardness from 430 to 800 HV. The outstanding performance of $NiCo_{0.6}Fe_{0.2}CrSiAlTi_{0.2}$ coating deposited by PS is demonstrated as a capable overlay coating for high-temperature applications involving particulates.

APS was used to deposit double-layer ceramic (DCL) TBCs, as shown in Figure 7 with a topcoat of high entropy rare-earth zirconate (HE-REZ) and an inner layer of yttria-stabilized zirconia (YSZ) on superalloys [35]. Thermal stability tests of HE-REZ/YSZ DCL and LZ/YSZ DCL (i.e., reference $La_2Zr_2O_7$ (LZ)) coating were conducted in air at 1100 °C. The newly designed HE-REZ/YSZ DCL coating reported exhibiting improved thermal stability with a lifetime of 53 times compared with that of 10 for LZ/YSZ DCL coating. Figure 8

shows photographs of both coatings before and after thermal cycling. The extraordinary performance of HE-REZ/YSZ DCL coating is attributed to sluggish kinetic diffusion as well as an improved match of coefficient of thermal expansion (CTE) of HE-REZ coating with the YSZ inner layer. In addition, HE-REZ ceramic coating retained its fluorite structure due to the mechanism as mentioned above, while the $La_2Zr_2O_7$ coating phase changed from fluorite to pyrochlore after the thermal cycling test.

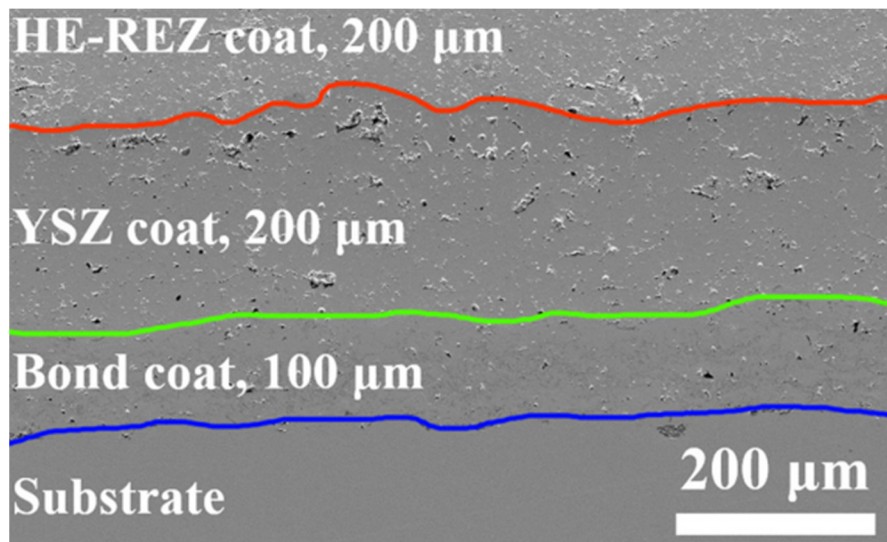

**Figure 7.** SEM image of the deposited double layer. Reprinted with permission from [35] Copyright 2020 Elsevier.

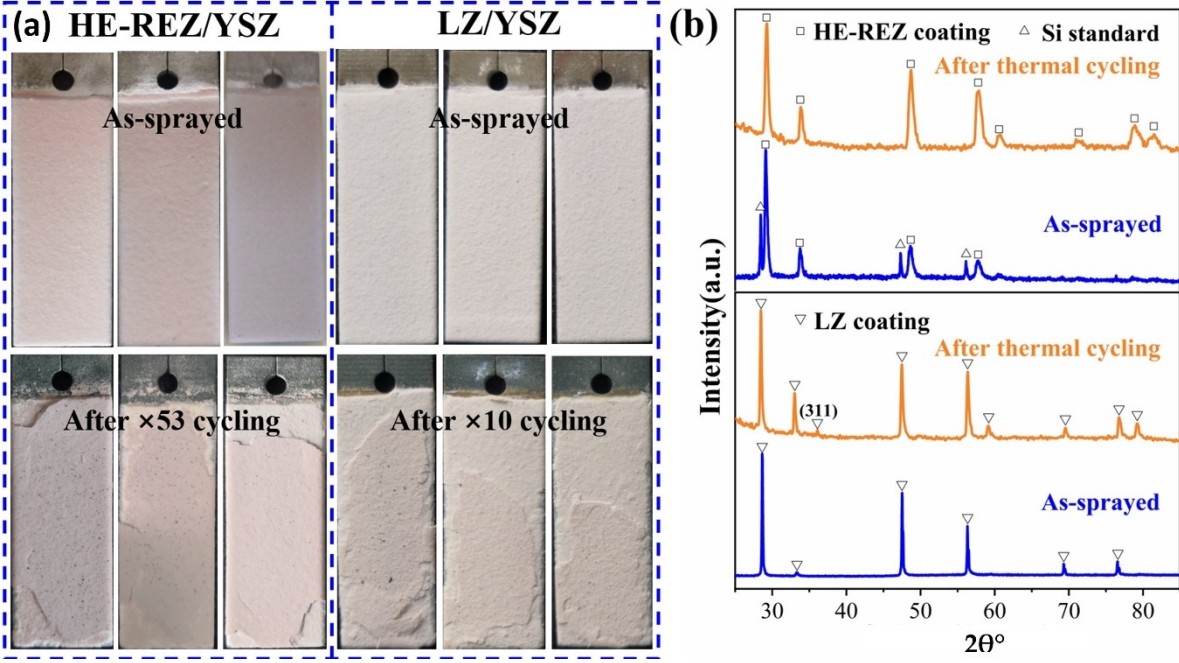

**Figure 8.** (**a**) Photographs and (**b**) XRD patterns of the as-sprayed coating and after thermal cycling test. Reprinted with permission from [35] Copyright 2020 Elsevier.

### 3.2.2. High-Velocity Oxygen-Fuel Spraying

High-velocity oxygen-fuel (HVOF) spraying, developed in the 1980s, is currently widely used to fabricate thick, dense, lower oxide contents, and high-strength bonding coatings [122]. A pressurized chamber (up to 0.8–0.9 MPa) is used for combustion, which is followed by a Laval-type barrel—the geometry of the barrel and hot gas expansion results

to generate supersonic velocities (up to 2000 m/s). A feedstock is introduced axially in the combustion chamber through a pressurized powder feeder [122]. HVOF coatings were investigated focusing on the characterisation of microstructural evolution, phase, wear, and oxidation behaviour [21,132,133].

Microstructure, mechanical, and anti-oxidation properties of HVOF-sprayed HEA ($Ni_{0.2}Co_{0.6}Fe_{0.2}CrSi_{0.2}CrTi_{0.2}$) coatings [21] were studied and compared with coatings produced from APS. The HVOF as-sprayed coating showed a structure of thick lamellae with low oxidation owing to the lower spraying temperature and higher output power, in comparison to APS coatings. The HVOF-deposited HEA coatings outperformed the MCrAlY coating at 1100 °C in terms of oxidation resistance. The initial weight gain of the coating as shown in Figure 9a was higher at 1100 °C than that of MCrAlY and is ascribed to the presence of Ti in HEA coatings, which oxidised quickly to form $TiO_2$, and weight gain became steady after 50 h. A cross-sectional view of HEA coatings at 1100 °C and 50 h test is shown in Figure 9b,c exhibiting a two-layer oxide scale. The external grey layer is made of Al-Ti-Cr mixed oxide added to initial weight gain. The dark thin layer is comprised of $\alpha$-$Al_2O_3$, which could limit oxygen diffusion and inhibit further oxidation at high temperatures. The thermally deposited HEA coating was described as an excellent overlay coating for high-temperature applications. However, coatings for such applications remain to be evaluated for both mechanical and functional properties according to defined standards to validate its practical applications, e.g., topcoat and thermal cycling. A thick mixed oxide of Ti-Cr-Al and $\alpha$-$Al_2O_3$ layers is required to prevent further oxidation.

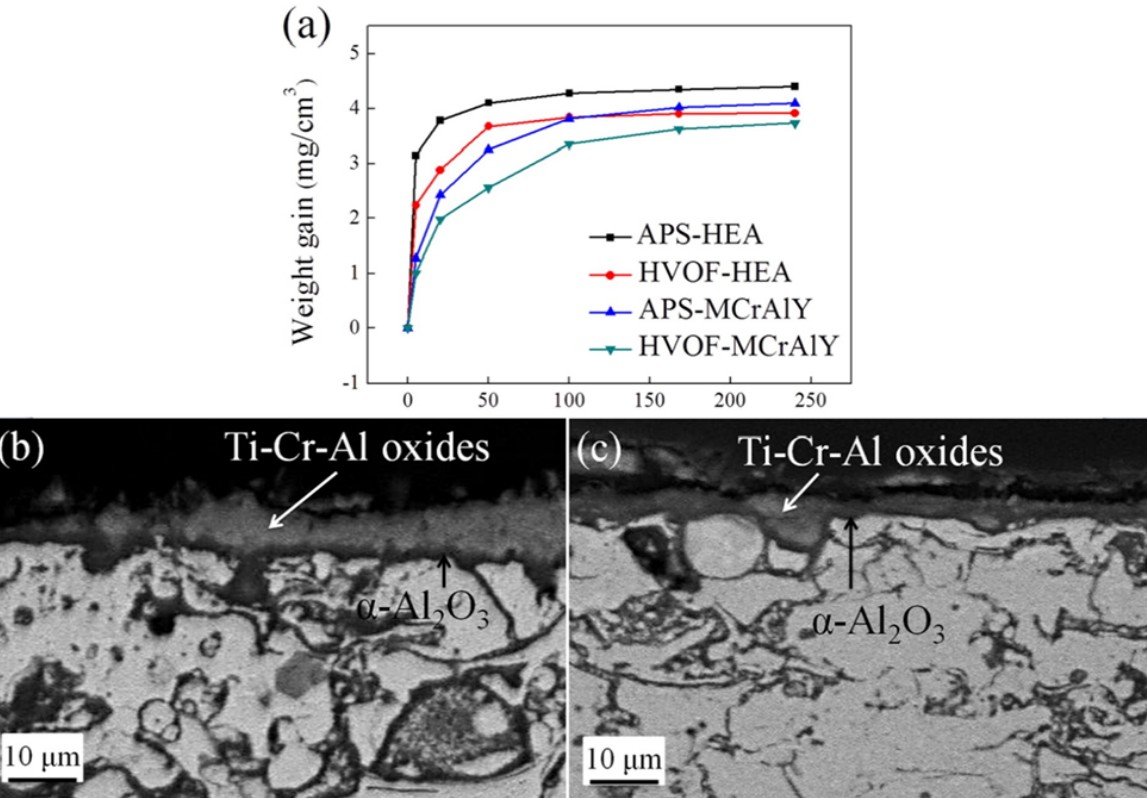

**Figure 9.** Shows kinetics of isothermal oxidation of thermally sprayed $Ni_{0.2}Co_{0.6}Fe_{0.2}CrSi_{0.2}CrTi_{0.2}$ HEA coatings at 1100 °C. (**a**) weight gain curve, (**b**) APS-HEA coating cross-sectional view, and (**c**) HVOF-HEA coating. Reprinted with permission from [21] Copyright 2017 Elsevier.

The $Al_{0.6}TiCrFeCoNi$ HEA coating was produced by HVOF to study the microstructure and wear behaviour at different temperatures [132]. The as-sprayed coating had two BCC phases similar to powder, suggesting that phase stability was maintained during deposition. A pin-on-disc test showed that the wear behaviour changed significantly with temperature.

Figure 10a shows a friction coefficient (COF) at room temperatures (RT), 300, and 500 °C. The COF at 500 °C is significantly lowered due to tribo-reaction and the oxide layer formed, serving as a lubricant. According to the wear rates as shown in Figure 10b, the wear at RT was dominated by the abrasion mechanism investigated using CLSM (confocal laser scan microscopy) and SEM, while the role of fatigue wear increased with an increase in temperature.

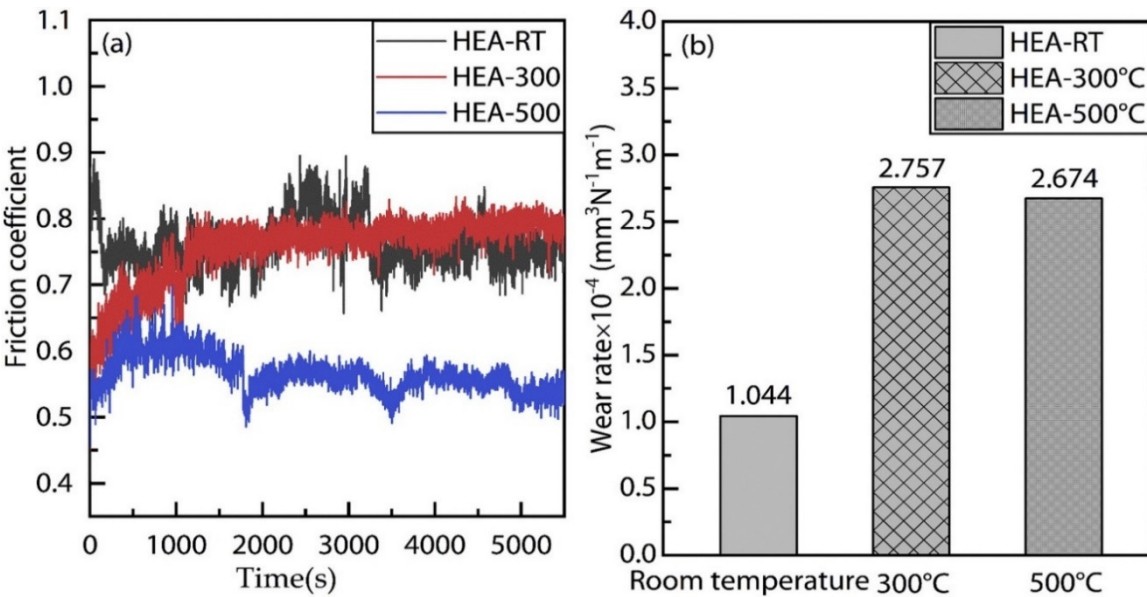

**Figure 10.** Pin-on-disk tests at various test temperatures: (**a**) friction co-efficient; (**b**) wear rates. Reprinted with permission from [132] Copyright 2019 Elsevier.

### 3.2.3. Cold Spraying

Cold spray is a newly developed solid-state coating deposition procedure, which is primarily a kinetic spray process using supersonic jets of compressed gas accelerating powder particles, at temperature well below particle melting temperature to extreme-high velocities (up to 1500 m/s) [122], thus no phase transformation and oxidation. However, there exist a considerable number of residual stresses [134–136]. In cold spray, the non-melted metals and metallic composites particles traveling at speeds between 500 and 1500 m/s plastically deform and consolidate on the substrate or underlying layer with an impact creating coating [137,138]. As shown in Figure 11, cold spray uses (i) convergent–divergent Laval nozzles upstream pressure of 2–2.5 MPa, while for a typical nozzle, the internal diameter is in the range of 2–3 mm. (ii) He, $N_2$ or a mixture of both is used as a carrier gas for deposition at a very high flow rate (up to 5 $m^3$/min). The mass flow rate of the gas must be kept 10 times the entrained feedstock (powder), the carrier gases are pre-heated up to 600–700 °C to prevent liquefaction under expansion and increased velocity.

The isothermal oxidation of AlCoCrFeNi HEA coating was explored in the TBC system after being deposited through cold spray for its potential application as a bond coat [29]. The coating was produced at 400 °C and 10 bar pressures on a Ni-superalloy. It was found that due to the characteristic features of cold spraying, the as-sprayed coating retained the mechanical alloying phases, with less inter-splat porosity and better interparticle bonding. A protective alumina layer was formed after isothermal oxidation of the deposited coating at 1100 °C for 25 h. The coating also experienced some internal oxidation possibly due to an extensive network of grain boundaries in the nano-crystalline powder and porosity in the coating. In addition, the dissolution of Mo from the substrate was noticed at the coating–substrate interface, suggesting high kinetic diffusion. This study was intended to develop an oxidation-resistant HEA coating for high-temperature application and elucidate the possible factors governing oxidation mechanisms. The cold spray was demonstrated as a viable potential coating deposition technique for HECs. However, the authors in this

study stopped to conduct a long-term oxidation test, thermal cycling to confirm adhesive strength, thermal shock resistance, and thermal expansion mismatch [129]. The cold state deposition nature of the cold spray of the deposited materials loses ductility also due to its nature the probability of voids and porosity in the deposited coating, which is higher [137].

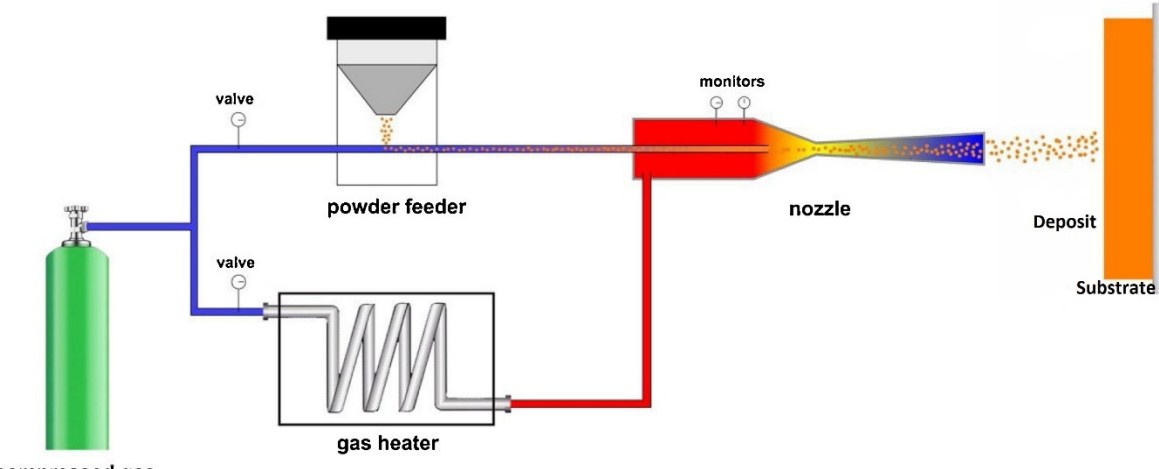

**Figure 11.** A schematic of the cold spray process. Reprinted with permission from [139] Copyright 2019 Elsevier.

In conclusion, the different variants that come under the general name of thermal spraying, including PS, HVOF, and cold spraying, had performed outstandingly in high-performance coating deposition in the aerospace and nuclear sector. The properties of the coating can be tailored through fine-tuning of the TS process parameters. Thermal-sprayed HEA coatings exhibit characteristic-type lamellar structures with weak lamellar bonding, defects (pores and micro-cracks), inhomogeneous, and anisotropic structures, which need to be further enhanced via optimizing process parameters.

### 3.2.4. HEAs Feedstock Synthesis

The final microstructure and properties of coatings not only depend on the TS process parameters but also the synthesis routes of feedstock. Four synthesis ways have been identified and used to prepare HEMs feedstock for various deposition techniques mainly for thermal spraying: (1) Gas Atomisation, (2) Arc Melting followed by Mechanical Milling, (3) Mechanical Alloying, and (4) Blending.

### I.    Gas Atomisation

The gas atomisation is one of the favoured routes for HEMs feedstock preparation fulfilling the condition of high mixing entropy. In HECs, community feedstock through inert gas atomisation has gained popularity due to its uniform composition. In this process, a liquid alloy is forced under pressure through a nozzle in an inert gas atmosphere. The spherical droplets formed from the liquid stream quickly solidify [140]. Due to the faster cooling rate in this process, phase separation occurs at very fine scales, as shown in Figure 12a,b [139]. The greater acceptance of HEA feedstock through gas atomisation is because the particles produced are spherical, having high flowability and homogeneity. Produced particles of a wide size range are sieved to the desired cut size.

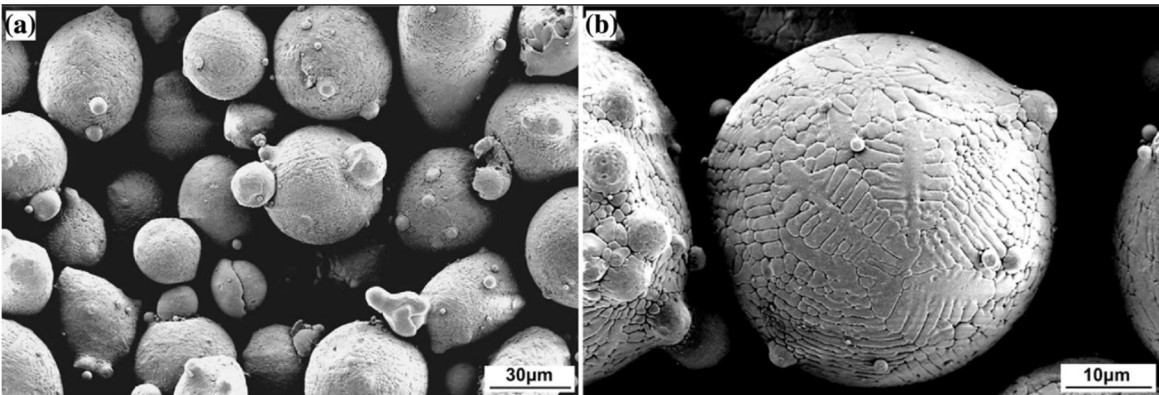

**Figure 12.** SEM image of HEA particles produced through gas atomisation (**a**) spherical morphology and satellite particles, and (**b**) dendritic structure visible at high magnification. Reprinted with permission from [139] Copyright 2019 Elsevier.

### II.　Arc Melting Followed by Mechanical Milling (AM-MM)

Arc melting is one of the popular HEMs liquid processing routes. In this method, selected alloy composition is melted in an arc-melting furnace with torch temperature (>3000 °C) that can be controlled by altering the torch power. Upon cooling, the alloy buttons are crushed for size reduction, followed by ball milling into powders. Arc melting is important to achieve the concept of configurational entropy in the liquid form of the alloying elements. To have a feedstock for thermal spray, pulverisation is required, which is size reduction. The size reduction process and mechanism of an arc melted alloy are shown in Figure 13. HEA ($Ni_{1.5}Co_{1.5}CrFeTi_{0.5}$) was processed by three different methods (AM-MA, Mechanical Alloying and Blending) [141]; the relating XRD patterns after each method are shown in Figure 14a–c. The arc melting XRD analysis of the five elements (Ni, Co, Cr, Fe, and Ti) is shown in Figure 14a. It was found that three phases present after mechanical alloying transformed to a single-phase FCC solid solution with a 3.75 Å lattice parameter. Such a transformation is because of a thermodynamic drive of the material to achieve the most stable state attainable in a condition [142].

Data adopted concerning the milling parameters, such as rotational speed, milling duration, media used, and the amount of feed charged, are not always provided. In ball milling, the particle produced is irregular in shape since the particles are split fragments of the casted buttons. Hence, the irregular shape particles produced adversely influence powder flowability, its adverse effect is minimised by adjustment of the spray parameters. A relatively homogeneous microstructure in terms of phases is observed in these powders as alloy formation has already occurred during the casting procedure.

### III.　Mechanical Alloying (MA)

MA is one of the well-known HEAs processing routes [144]. In this solid-state processing route, powder feedstock is produced by constant cold welding, fracturing, and re-welding in an energy-intensive ball-milling process [144]. Powders of different elements are subjected to high-energy impacts during high-speed rotation, causing them to undergo repeated cold welding and fracture, to achieve an atomic-scale mixture [144]. Over 200 reports reviewed in [145] suggest that the phase formed is dependent on milling process parameters (i.e., milling duration, milling atmosphere, milling medium, process control agent, etc.). All these main parameters affect alloying quality, indicating that there is no unique recipe for producing HEA via MA, unlike arc fusion. In Figure 14b, XRD analysis showed the formation of three phases: the BCC phase had a lattice parameter of 2.88 Å (i.e., nearly Fe and Cr) and two FCC phases had lattice parameters of 3.52 and 3.58 Å, respectively (i.e., both nearly Ni).

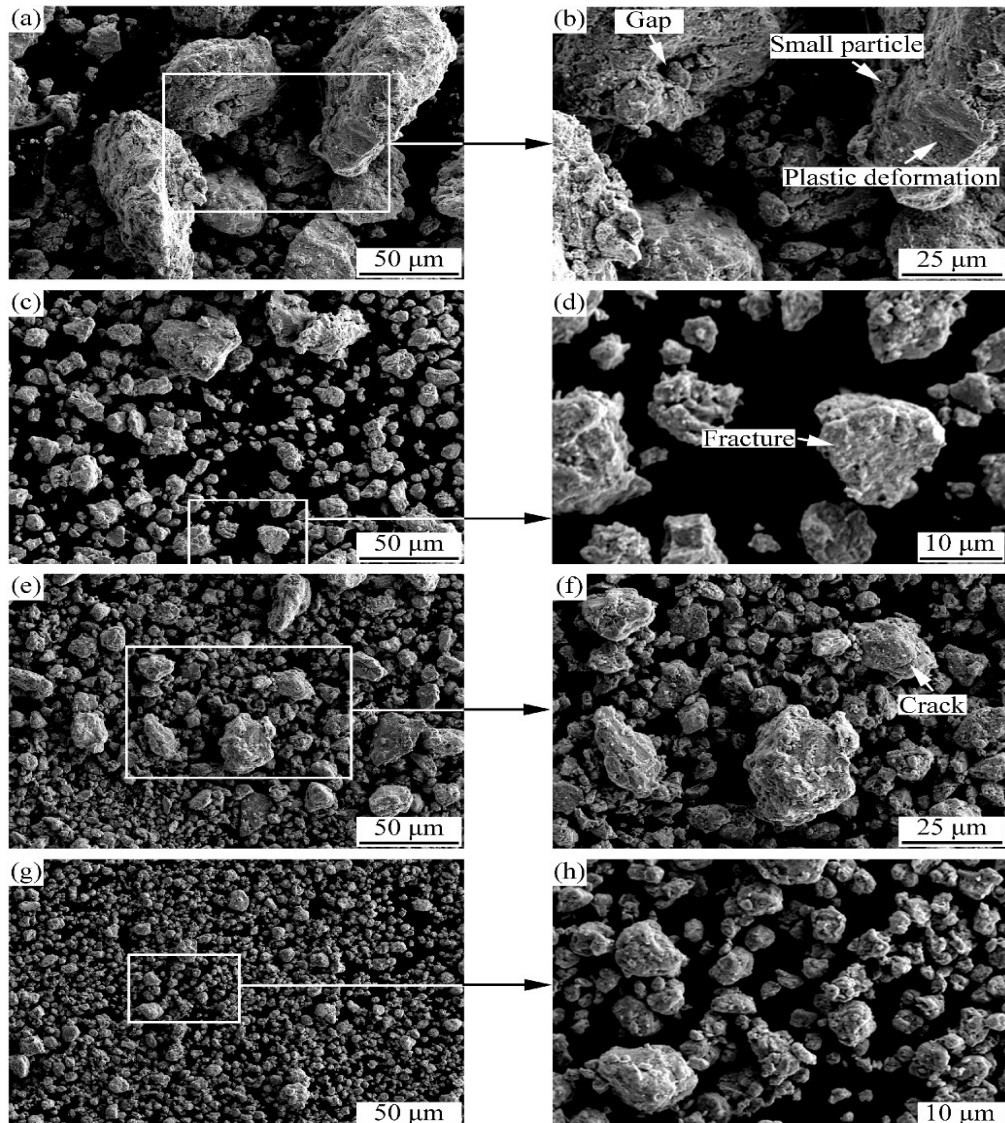

**Figure 13.** Size reduction processes arc melted AlCoCrFeNiSi alloy at different milling times: (**a**,**b**) 5 h; (**c**,**d**) 10 h; (**e**,**f**) 20 h; (**g**,**h**) 30 h. Reprinted with permission from [143] Copyright 2019 Elsevier.

## IV. Blending

Blending is a mixing of powders without promoting any alloying or bond formation [146]. For feedstock, homogeneous mixing is crucial, but in this route, the particles of the mixed blend retain their shape and size and other elemental qualities. The XRD spectrum for blended-only feedstock for the design of HEA ($Ni_{1.5}Co_{1.5}CrFeTi_{0.5}$) in Figure 14c shows clear peaks of the five respective individual elements in the mixture.

Blending is therefore effective for mixing pre-alloyed powders but is not a recommended way to produce HEA feedstock, as it is unable to fulfil the fundamental philosophy of configural entropy. A high configurational entropy demands a discrete representation of constituent elements in a lattice.

In a nutshell, gas atomisation is tailor-made for synthesising HEMs and is suitable for preparing feedstock required for coating processes. A strong alternative to it is mechanical alloying, but that requires the optimisation of milling parameters, which is time and energy intensive. A preferred technique for preparing lab-scale HEA feedstock is arc melting followed by mechanical milling, while blending is good for improving feedstock properties in post-alloying. As opposed to pre-alloyed feedstock for TS, in laser cladding, coating is also achieved with blended elemental powders. In some research studies, HEA coating

through laser cladding is produced with powders synthesised via mechanical alloying [147] or gas atomisation [147] rather than utilising elemental blends.

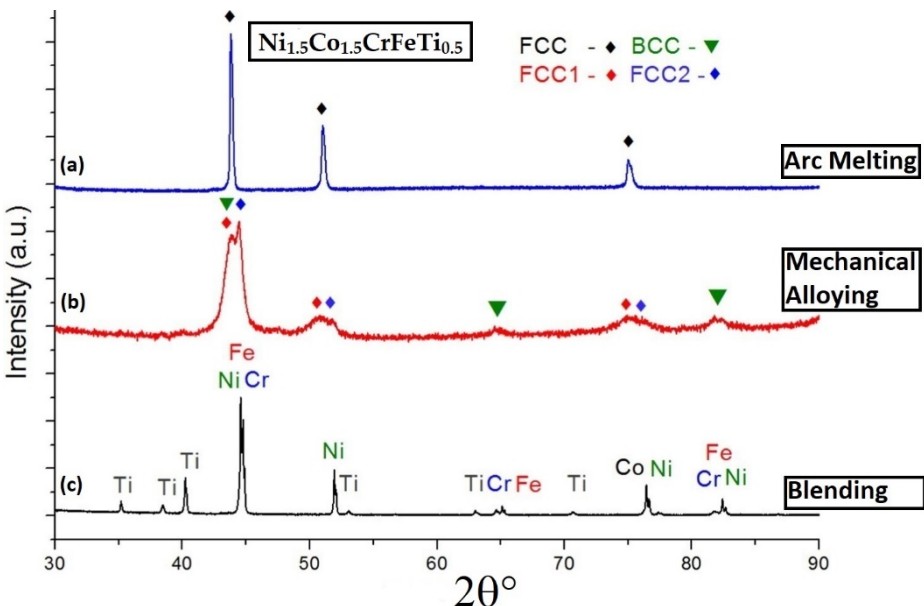

**Figure 14.** XRD patterns of (**a**) blended powder feedstock, (**b**) mechanical alloyed powder, (**c**) Spark Plasma Sintering (SPS) compacted alloyed powder. Reprinted with permission from [141] Copyright 2020.

### 3.3. Vapour Deposition Methods

The general terms PVD and CVD are techniques broadly used for thin-film coatings. The target material with the help of an external high-energy source (including arc, resistance heating, or ion bombardments) is transformed as a thin film on the substrate surface. Currently, PVD and CVD methods have been used for conducting vast research, for instance magnetron sputtering, reactive magnetron-sputtering, and vacuum arc deposition to produce HECs with needed surface properties [148,149]. A diagram shown in Figure 15 exhibits the magnetron sputtering process of CoCrFeNiMn thin film on substrate. High-entropy nitrides, carbides, and carbonitrides are the most common type of HECs fabricated by vapour deposition methods. These deposited coatings had a common characteristic: columnar structures, FCC/BCC solid solution or amorphous phases, and ultra-high hardness. Nevertheless, high-entropy coatings/films thickness produced by vapour deposition is often of submicron, as opposed to laser cladding, TS, and related deposition processes. These HECs also tend to form nano-sized structures, leading to substantial improvement in physical and mechanical properties in terms of wear, thermal stability, oxidation, and irradiation resistance [150]. For instance, the high entropy nitride coatings due to their higher diffusion resistance are used as diffusion barriers in integrated circuits [151]. The high-entropy nitride coatings formed are primarily based on thermodynamically stable nitride forming elements (i.e., Zr, Ti, V, Hf, etc.) [152]. In addition to composition, the deposition environment such as $N_2$ pressure, voltage bias, and other factors also have striking effects on the surface morphology and phase structure of the high-entropy nitride coatings [153].

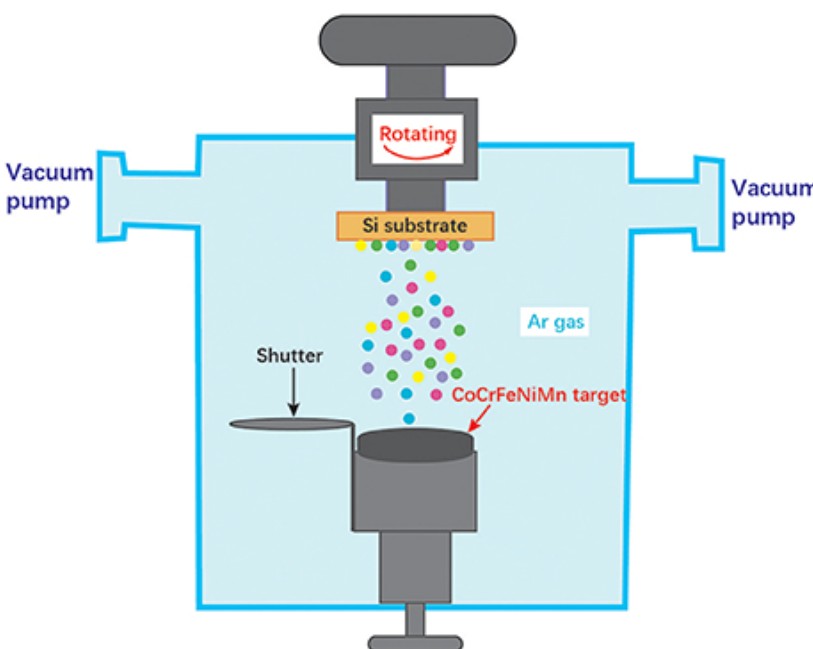

**Figure 15.** A schematic of multi-target magnetron sputtering for CoCrFeNiMn thin film on substrate. Reprinted with permission from [154] Copyright 2020 Elsevier.

### 3.3.1. Magnetron Sputtering

Magnetron sputtering is a broadly used technique for HEA film deposition, based on its characteristic features such as versatility in controlling film composition and properties through varying deposition parameters, such as substrate temperature, substrate bias, sputtering power and atmosphere ($N_2$, Ar, $CH_4$ and/or $O_2$) [152,155,156]. Carbide, nitride and carbonitride-based high-entropy ceramic coatings/films produced through magnetron sputtering have gained substantial interest as protective coatings. Transition metals HEAs form thermodynamically stable nitride/carbides compounds that vary along the groups in the periodic table (see Figure 2) [65]. The ultra-thin films with excellent physical and mechanical properties, of ultra-high hardness [152,153], wear resistance, [148,157] thermal stability, anti-oxidation [149], anti-corrosion [158], radiation resistance [159] and diffusion barriers [160] can be produced with magnetron sputtering.

Zirconium alloys have low neutron absorption, higher corrosion and wear resistance and are commonly used as nuclear cladding material. However, in conditions of loose-of-coolant accident, Zr alloy reacts with water, producing a large volume of $H_2$ gas leading to an explosion (also radioactive material leakage). Therefore, a study was conducted [27] to investigate the microstructure, mechanical properties, and wettability for AlTiCrNiTa HEA thin films deposited on Zr-4 through magnetron sputtering. XRD and TEM analysis revealed that the HEA coating had an amorphous structure with minor FCC nano-crystallised domains. The mechanical properties assessment showed that the coating had strong adhesion, suggesting that AlTiCrNiTa coating adheres well to the Zr-4 substrate. The deposited coating had a high hardness of 18.34 GPa, exhibiting good wear resistance. For analysing corrosion properties, a 45-days autoclave test at 320 °C confirmed that the AlTiCrNiTa HEA coating is effective, but a weight loss occurred during the test conditions. As the author claims that no Zr oxides formed, which suggests that the coating has excellent corrosion resistance properties. However, weight loss during the corrosion resistance experiment shows that no protective layer formed, and the corroded layer went away with time. Long-time exposure will eventually expose the substrate material with gradual and continuous removal of the corroded layer. Pre-oxidation treatment before autoclave testing may help improve the corrosion resistance with the formation of a protective oxide layer.

### 3.3.2. Vacuum-Arc Deposition

Vacuum-arc deposition is another technique used for thin-film coatings and has been used for various high entropy-based films deposition utilising the arc heat energy to vaporise the target and deposit it on the substrate. The vacuum-arc deposition was used to study the effect of $N_2$ pressure [78] on the crystalline size, phase constitution, and hardness of (CrTiZrNbAlY)N HECe nano-composite coatings, but these have not been tested at high temperature. Overall, magnetron sputtering, and vacuum-arc deposition techniques are used for the fabrication of high-entropy-based ultra-thin films of submicron thickness, in particular the HECe films [161]. The films deposited are mostly metal nitrides/carbides binaries having an FCC structure of NaCl-type in which Na sites are occupied by the metallic elements, while the Cl sites are occupied by N [65]. Such HECe films of characteristics columnar, nanocomposites, and amorphous structures still hold features and properties of HEMs, such as low residual stress, thermal stability, high wear, and oxidation resistance. In addition, under the four core effects of HEMs, the mechanical and functional properties of these HECe films are exceptional by designing appropriate composition and tailoring fabrication parameters, including the reactive gas flow ratio.

In summary, the equipment of these coating depositions is relatively costly and is mostly used to produce thin films. The thin films on nm scale are produced and are always challenging to evaluate for mechanical properties. These techniques are very much versatile and can be used for low friction, wear resistance, anti-bacterial, and fuel cell coatings.

## 4. Properties of High-Entropy Coatings

High-entropy coatings' unique mechanical properties, i.e., ultra-high hardness and wear resistance, have been demonstrated in many studies [162] with potential use in many engineering applications as protective hard coatings. Apart from mechanical aspects, functional properties are equally important for many applications, e.g., corrosion, oxidation, radiation, diffusion resistance, etc. This section summarises the main functional properties of HECs and analyses their fundamental mechanisms.

### 4.1. Corrosion Resistance

The performance and durability of the substrate material can be enhanced by the application of surface coating. It is more adoptable to exploit anti-corrosion properties of bulk HEAs as coating for the corrosion protection of substrates. Bulk HEAs containing elements such as Cr, Co, Ni, and Ti are having enhanced corrosion resistance properties [163,164]. In addition, it has been reported that Mo (<3%) addition increases the amount of corrosion resistance oxides ($Cr_2O_3$) in the passive film that enhances HEAs pitting corrosion resistance in an aqueous solution containing chloride [165]. In some cases, the addition of certain elements showed adverse effects on the corrosion resistance due to the inevitable changes in microstructure, elemental segregation, or intermetallic formation. For instance, the Al addition to FeCoCrNi HEAs promotes the formation of the BCC intermetallic phase that is Al-Ni-rich and Cr-depleted [166]. On the other hand, Cu and Mn elements are also labelled as having negative effects on anti-corrosion [167,168].

The corrosion behaviour has been studied for the CoCrFeNiW and $CoCrFeNiW_{0.5}Mo_{0.5}$ HEAs coating, which was prepared by mechanical alloying and vacuum hot-pressing sintering [165]. A comparison of corrosion behaviour is shown in Figure 16 in the form of potentiodynamic curves in 3.5 wt % NaCl solution for CoCrFeNiW, $CoCrFeNiW_{0.5}Mo_{0.5}$ and substrate. Both HEA coatings have high positive corrosion potential ($E_{corr}$) and lower corrosion current density ($I_{corr}$) in comparison to the substrate, suggesting high corrosion resistance because the wider the passive region $\Delta E_P$ ($E_{corr} - E_{pit}$) is, the better the corrosion resistance. Mo addition enabled the coating to have a smaller corrosion current density, higher pitting potential, and a wider passive region. Therefore, a secondary pseudo-passive plateau appeared in the potentiodynamic polarisation curves as shown by the Figure 16 inset, signifying that Mo addition increased the pitting resistance of the HEA coatings.

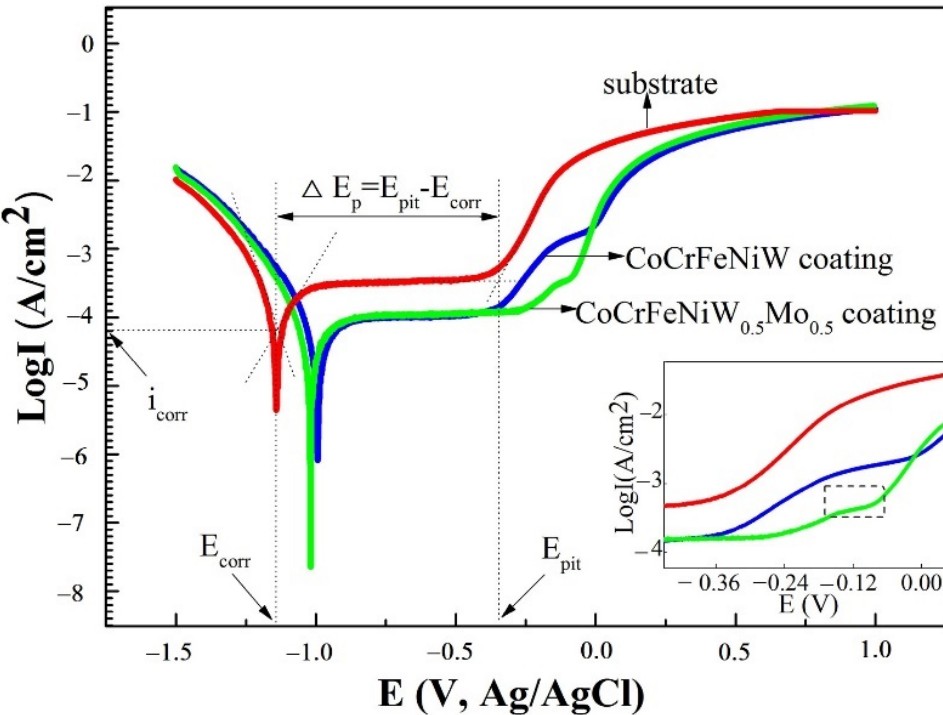

**Figure 16.** Potentiodynamic polarisation curves for the Q235 steel and vacuum hot pressing sintered-HEA coatings in 3.5% NaCl solution. Reprinted with permission from [165] Copyright 2016 Elsevier.

The positive effect of the Ti element for improving corrosion resistance was found in laser-cladded $Al_2CrFeNiCoCuTi_x$ HEA coatings due to the formation of FCC, BCC and Laves phases [169]. The impact of Ti addition on corrosion resistance was determined by potentiodynamic polarisation curves in 0.5 mol/L $HNO_3$ solutions for both coatings and substrates. Compared with the substrate (i.e., Q235 steel), the free corrosion current density for the coating was more positive and reduced by one to two orders of magnitude, indicating that the $Al_2CrFeNiCoCuTi_x$ coating played a protective role in the $HNO_3$ environment. With higher Ti content, the corrosion resistance against $HNO_3$ was enhanced and $Ti_{0.5}$ showed the best corrosion resistance among all. This is mainly because the addition of Ti can promote the formation of a dense protective oxide film in oxidising acid and the film remains stable in a passive state.

AlMoCrNbZr/(AlMoCrNbZr)N multilayer coatings [170] (the thickness varied from 5 to 50 nm) were deposited on an N36 alloy through magnetron co-sputtering to enhance the corrosion resistance in the aftermath of a loss-of-coolant accident. XRD and TEM revealed a multi-layered structure consisting of FCC (AlMoCrNbZr)N and an amorphous AlMoCrNbZr layer. The results indicated that the corrosion resistance of the N36 alloy tripled in static water at 18.7 MPa and 360 °C for 30 days due to multi-phase oxides ($Nb_2Zr_6O_{17}$, $ZrO_2$ and $Cr_2O_3$) formed on the surface. A multi-layer AlMoCrNbZr/(AlMoCrNbZr)N coating is useful in inhibiting Al diffusion and boehmite phase formation during autoclave corrosion of an AlMoCrNbZr alloy. The AlMoCrNbZr/(AlMoCrNbZr)N multi-layer coating of a 50 nm layer step with demonstrated better protective properties (Figure 17) in comparison to 5/5 nm and 10/10 nm multi-layers and single-layer AlMoCrNbZr.

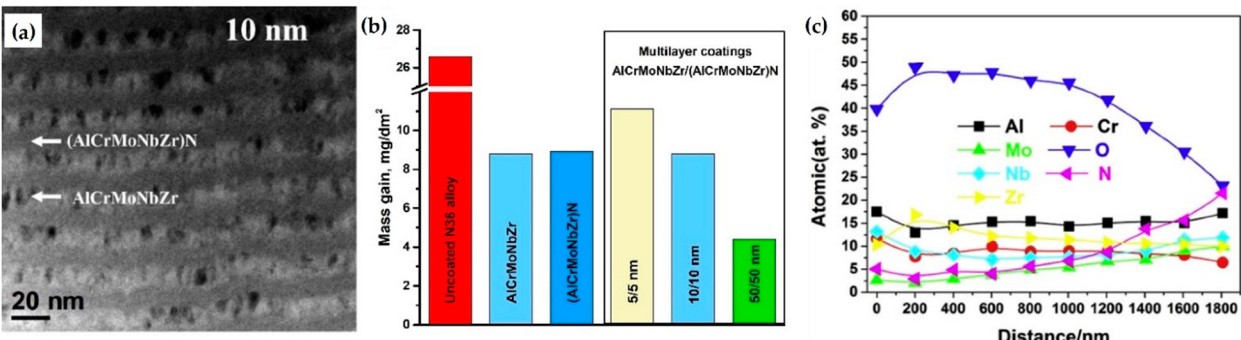

**Figure 17.** AlMoCrNbZr/(AlMoCrNbZr)N HECe coatings: (**a**) TEM image of the cross-section of 10/10 nm coatings, (**b**) HECe-coated N36 Zr alloy mass gains during autoclave test at 360 °C, 18.7 MPa for 30 days. (**c**) Depth of elemental distribution of 50/50 nm HECe coatings after autoclave test [170]. Reprinted with permission from [171] Copyright 2018 Elsevier.

Apart from the inclusion of corrosion resistance elements in HEMs, the corrosion resistance in these materials requires investigation regarding four core effects to establish its impact on its anti-corrosion properties. For instance, the influence of a disordered structure on the corrosion properties of HEAs has been neglected, and most of the research is on corrosion mechanisms in aqueous solution, which is mainly related to the composition and microstructure. Despite the inherited chemical properties of coating materials, corrosion resistance is also dictated by the fabrication procedure giving different microstructure and phase constitution. Therefore, it is necessary to make a clear correlation among the composition, microstructure, deposition technique, and corrosion resistance of HEMs to lay the theoretical foundation for the design and expansion of corrosion-resistant HEC materials. Research on corrosion behaviour and mechanisms of the HECs materials require the use of electrochemical measurements coupled with advanced surface characterisation techniques such as transmission electron microscopy, X-ray photoelectron spectroscopy (XPS), Auger electron spectroscopy (AES), and time-of-flight secondary ion mass spectroscopy (ToF-SIMS). In this way, the composition, microstructure, and coating properties of HEAs can be analysed in detail to reveal a corrosion-resistant mechanism.

### 4.2. Oxidation Behaviour

One of the most critical requirements is its ability is to maintain its surface stability in an oxidizing high-temperature environment. High-entropy alloys are alternatives for anti-oxidation property at high temperature because of their special structure and properties. Recently, HEAs have been developed exhibiting enhanced oxidation resistance properties at high temperature. The better oxidation resistance of these alloys is attributed to sluggish kinetic diffusion and the presence of specific elements such as Al, Cr and Si having a dual effect of lower oxygen diffusivity and denser protective oxide layer formation. Most of the HEAs investigated for oxidation behaviour are BBC-HEAs incorporating Al and Si; both act as BCC stabilisers and form the protective $Al_2O_3$ and $SiO_2$ layers that favour the oxidation properties. The section below reviews some of the work completed on the application of HEAs as coating material for high-temperature application.

In TBCs, MCrAlY (M = Ni/Co, or both) alloy is widely used as a bond coat for oxidation resistance, which is mainly achieved by the formation of continuous, adherent, and gradual growth of $Al_2O_3$ scale on the coating surface at high temperature [172,173]. It has been investigated that higher oxidation rates of MCrAlY at/above 1100 °C cause premature failure of the $Al_2O_3$ scale [174–176]. A recent report on Y-Hf doped AlCoCrFeNi HEA [28] used as a bond coat investigated its oxidation resistance and strength at/above 1100 °C, as well as chemical compatibility with nickel superalloy substrate. For comparison and investigation of the oxidation behaviour of Y-Hf co-doped AlCoCrFeNi HEA and CoCrAlNi alloy (CAN), both coating samples were produced with the help of the SPS

process on the nickel superalloy at a sintering temperature of 1050 °C. In HEA coating, a characteristic two-phase nanostructured coating comprising A2 and β phases was observed. Isothermal oxidation test analysis of both types of the coating was conducted at 1100 °C in a furnace in an atmospheric environment. Figure 18 illustrates the evolution of oxide thickness for two coating types at 1100 °C. An extremely low oxidation rate having an oxidation rate constant of $1.3 \times 10^{-2}$ µm$^2$/h at 1100 °C displayed by HEA coating—50% lower than ($2.6 \times 10^{-2}$ µm$^2$/h) CAN-coating. The low oxidation rate of this HEA coating is associated with a predominant higher width of columnar grains in the $Al_2O_3$ scale in comparison to CNA coating [177,178]. The scale/metal interface wrinkling is well controlled and is linked to the higher creep resistance of HECs, originating from the sluggish diffusion effect in HEMs. The HEA coating showed superior oxidation resistance than that of CNA coating with no spallation of oxide scale after 1000 h oxidation at 1100 °C.

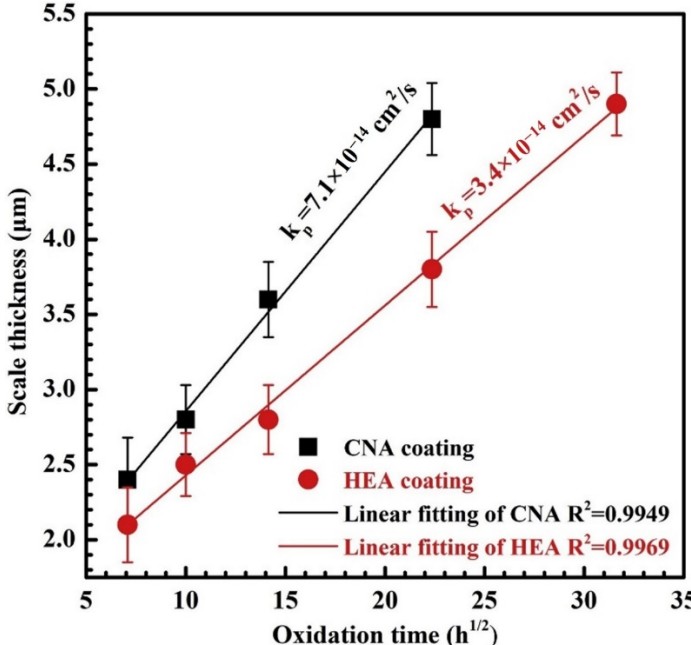

**Figure 18.** Oxide scale evolution for two coating types at 1100 °C as a function of the square root of oxidation time. Oxide scale thickness on the CNA coating was merely measured until 500 h oxidation. Reprinted with permission from [28] Copyright 2021 Elsevier.

The oxidation resistance and Al diffusion behaviour of $CuAl_xNiCrFe$ HEC deposited on superalloy through high-speed laser cladding to evaluate its suitability as a bond coat in TBCs [24]. In TBCs, the critical factor that leads to spallation and eventual failure is the TGO layer is the rapid growth caused by $O_2$ high ionic diffusivity and top layer inter-connected pores [175,179–181]. Therefore, to improve component service life, it is critical to control the TGO growth rate [182,183]. The oxidation behaviour of the four different $CuAl_xNiCrFe$ (x = 0.5, 1, 1.5, 2) HEA bond coats was assessed at high temperature. The microstructure of the bond coats showed transition from columnar to equiaxed grain with an increase in Al concentration. The bond coats were tested in harsh experimental conditions at an isothermal temperature of 1100 °C for 50 and 100 h. The TGO layer on all the bond coats grew at a parabolic rate, as shown in Figure 19, indicating a similar growth rate of all the bond coats. The oxide layers formed in the first 50 h were of a single phase with no variation in oxide composition among different samples of the bond coat. A spinal structure was formed in TGO on $CuAl_{0.5}NiCrFe$ during 100 h due to Al depletion because of lower concentration and prolonged oxidation time, while the remaining three compositions showed no variation in the oxide layer. The alumina oxide layer on the surface of the $CuAl_xNiCrFe$ HEA bond coat exhibited the morphology of $\alpha$-$Al_2O_3$, which protected subsequent coating surface oxidation. The $CuAl_xNiCrFe$ bond coats oxidized at 1100 °C showed entropic stability and

anti-diffusion because a large elemental concentration gradient was maintained between the coating and the substrate. The authors suggested that $CuAl_xNiCrFe$ HEA coatings deposited by high-speed laser cladding can potentially be used in TBCs as a bond coat due to their lower volume shrinkage at high temperature and higher oxidation activation energy in comparison to traditional bond coats but only at the laboratory testing stage.

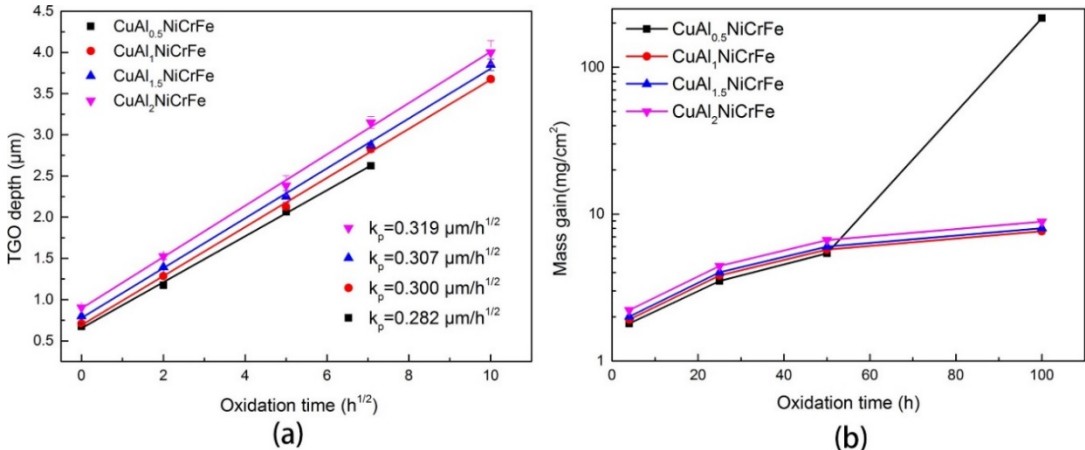

**Figure 19.** (**a**) shows TGO thickness evolution (**b**) mass gain at isothermal oxidation at 1100 °C of the $CuAl_xNiCrFe$ (x = 0.5, 1, 1.5, 2) bond coats. Reprinted with permission from [24] Copyright 2021 Elsevier.

The investigation of oxidation behaviour of HEA is a budding research topic and the oxidation behaviour of HEAs based on the FeCrCoNi alloy system is widely studied in the context of alloying elements effects. Studies on the oxidation behaviour of RHEAs report poor oxidation resistance due to the refractory element's oxygen affinity and weak protective oxide-scale formation. Generally, the oxidation behaviour of HEAs is greatly influenced by the chemical composition and nature of the oxide scale. The effects of separate alloying elements, alongside their combinations on protective complex oxides formation mechanisms, relevant elemental activities, and the rate of internal oxidation needs to be studied in detail. As in recent studies, encouraging results reported extremely low oxidation rates of HEAs co-doped with reactive elements. In addition, to obtaining insight and distinguishing these complex oxides, it is important to gain a better understanding of chemical bonding, atomic ordering, and oxides defect structures. The effect of the microstructure of the HECs coating should be studied in detail to predict their high-temperature oxidation behaviour. The grain size, chemical composition, volume fraction, and distribution of secondary phases may have large effects on the oxidation behaviour of the coatings produced from HEMs.

### 4.3. Radiation Resistance

Nuclear fission and fusion reactors require structural materials to have the reliability to tolerate irradiation [184]. Coatings are cladded on the surface of nuclear fuel rods with a selection of materials that hold excellent thermo-mechanical stability, high creep, corrosion resistance, and lower radioactive activation [185,186]. The atomic-scale ions radiation on these solid surfaces create atomic translations and thermal spikes. HEMs' intrinsic properties, including high entropy and high atomic-level stresses resulting from atomic radii variation of mixing elements, could promote amorphisation, local melting, and recrystallisation upon atoms/ions radiation and thermal spikes, thus creating lesser defects in comparison to conventional alloys [187,188]. These features make HEMs an excellent candidate as a next-generation nuclear coating material to be applied in high-radiation and high-temperature environments [170,189–191].

Komarov et al. [192] for the first time studied the influence of He$^+$ ions irradiation on a (V, Hf, Zr, Ti, Nb) N nanostructured coating deposited through vacuum arc deposition. He$^+$ ions of energy 500 keV irradiated with a fluence varying from $5 \times 10^{16}$ to

$3 \times 10^{17}$ ions/cm$^2$ on samples. Before irradiation, only one FCC phase exists with a grain size of 240–250 nm which was reduced to 50–60 nm after He$^+$ ions irradiation with a fluence of $(1–2) \times 10^{17}$ ions/cm$^2$. Meanwhile, higher fluence irradiation of $2 \times 10^{17}$ ions/cm$^2$ reduced grain size to 5–10 nm and confirmed that irradiation can make grains finer. The mechanical properties' stability after radiation was studied to evaluate its influence on coatings by conducting a hardness test that revealed a nonlinear relationship with fluence. At a low fluence rate (up to $1 \times 10^{17}$ ions/cm$^2$), the hardness increased by 4%–10%, but at higher fluence ($3 \times 10^{17}$ ions/cm$^2$), the hardness decreased by 9%–15%. However, in the coating, no new phase precipitation was observed nor did the composition of the coatings change upon irradiation up to a fluence of $2 \times 10^{17}$ ions/cm$^2$. Furthermore, no blistering in the coatings was observed in SEM analysis at any fluence rate, while for a fluence of $3 \times 10^{17}$ ions/cm$^2$, the film coating exfoliated at the point of radiation-induced defects where helium is maximal. It was found that a (Ti, Hf, Zr, V, Nb) N coating irradiated with 500 keV He$^+$ ions do not cause any noticeable change in the phase and structure of the coating, while moderate irradiations add to its hardness, making it radiation-resistant and capable for claddings of fuel rods in nuclear reactors. However, it was not mentioned how long the coating was exposed to the radiation, which questions the long-term stability of the coating materials. In addition, materials for safety-critical applications are required to undergo a series of mechanical tests at temperatures close to the real applications and cannot rely on just one mechanical property.

Overall, producing high-density defect sinks, such as secondary phase as well as grain boundaries, is an established technique for reduction in residual defects in irradiation-tolerant materials as shown in oxide-dispersion strengthened (ODS) steels [193], Zr alloys and SiC ceramics [171]. However, despite these successes, in high-temperature and severe radioactive environments, the nanostructures in the above materials are unstable [193]. Therefore, it is supposed that the HEMs complex randomized the arrangement of elements and the atomic-level local chemical environment; additionally, the higher site-to-site lattice distortions can efficiently reduce the mean free path of electrons, phonons and magnons [194–196]. To obtain a complete understanding of the fundamental control mechanism on improved radiation tolerance, by the development of defect clusters at elevated temperature and high irradiation doses, a set of HEMs with different alloying elements needs to be further investigated. Insight about the performance of produced HECs under high-dose irradiation at elevated temperatures will help to correlate intrinsic properties and defect dynamics for radiation-tolerant materials.

### 4.4. Diffusion Barriers

The excellent thermal stability and diffusion resistance of several HEMs coating and films make it possible to be used as a diffusion barrier especially for maintaining the robust diffusion resistance under elevated temperature and lower thickness [197–199]. The TBCs lifespan is controlled by maintaining a reservoir of Al in the form of MCrAlY or β-Ni (Pt)Al, which form a protective layer of Al$_2$O$_3$ on the surface. Loss of Al from the coatings is caused by the continuous growth of the Al$_2$O$_3$ scale during thermal cycling and by coating/substrate inter-diffusion. Al inward diffusion into alloy and alloying elements outward diffusion could lead to a decline in the mechanical properties of the substrate. To suppress the inter-diffusion which leads to the loss of Al and penetration of Ni and other elements such as Ti, and Ta from the substrate, an HEA coating was applied. A 4 μm thick AlCoCrNiMo HEA coating was deposited on a superalloy substrate through DC magnetron sputtering between the protective coating of NiAlHf and superalloy and compared to the NiAlHf coating [160]. For simplicity in the text, the samples NiAlHf/HEA/N5 and NiAlHf/N5 are indicated as NHN and NN. The inter-diffusion behaviour and coating resistance to high-temperature oxidation were investigated in an isothermal oxidation test at 1100 °C in air for both sample types. Figure 20a shows cross-sectional morphologies of the samples after 50 h and 100 h. The NN specimen exhibited a characteristic microstructure of the inter-diffusion zone (IDZ) and secondary reaction zone (SRZ) after 50 h exposure while after

100 h, it exhibited rod/needle-like and topologically close-packed (TCP) phases—believed to be unfavourable to the superalloy's mechanical properties. The precipitated phases in the SRZ region are shown in Figure 20b. None of the above-mentioned unwanted phases were observed in samples with HEA barrier coating after long-term oxidation at 1100 °C, as can be seen in Figure 20c,d Alumina layers were formed at the interface of NiAlHf/HEA and HEA/N5. The formed alumina layers are supposed to have exceptional resistance to the inter-diffusion of alloying elements due to their high chemical stability and lower diffusion coefficient [200]. The NiAlHf coating with added HEA diffusion barriers exhibited improved oxidation resistance. The results show that the diffusion barrier based on HEA can effectively restrain alloying elements inter-diffusion with substrate and can be attributed to HEAs' sluggish diffusion. Moreover, it was observed that the presence of Co in HEA also plays an important role to suppress the formation of SRZ and TCP precipitates in the superalloy.

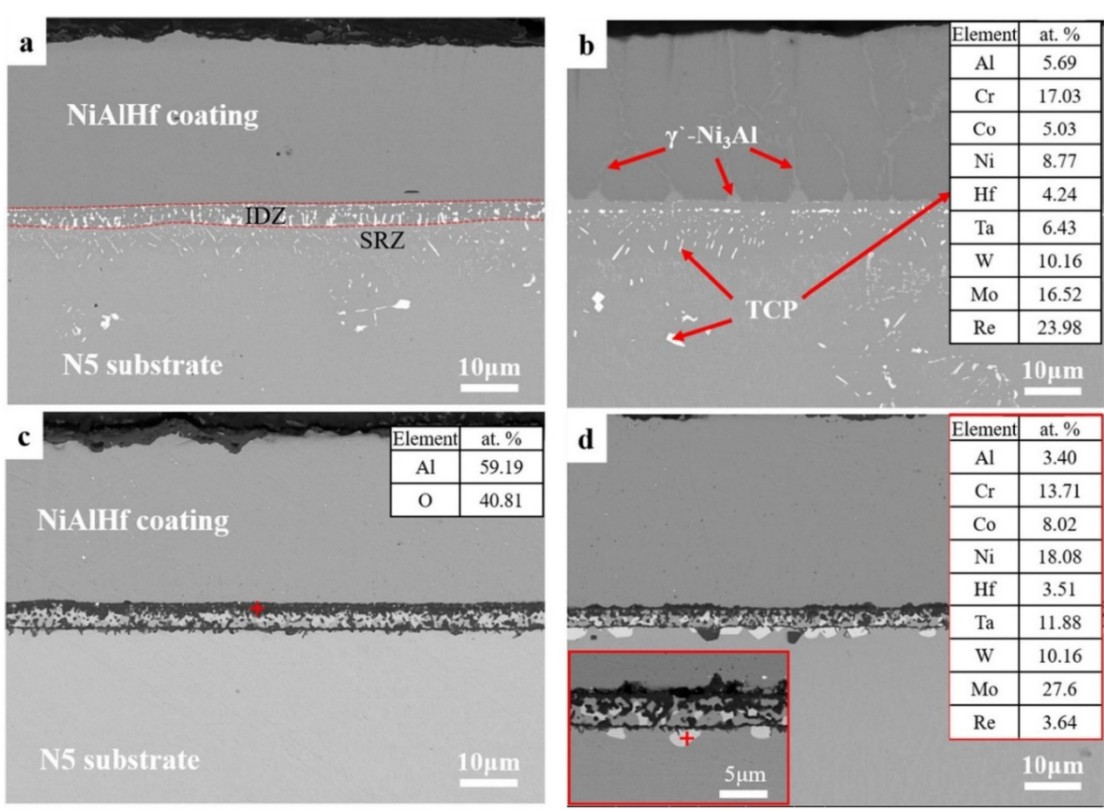

**Figure 20.** Specimen's cross-sectional morphologies after static isothermal oxidation at 1100 °C (Specimen NN after (**a**) 50 h and (**b**) 100 h oxidation, Specimen NHN after (**c**) 50 h and (**d**) 100 h oxidation). Insets of EDS of precipitates in NN and NHN specimens after 100 h oxidation. Reprinted with permission from [160] Copyright 2021 Elsevier.

Inhibiting rapid interdiffusion between Si and Cu in an integrated circuit, an efficient diffusional barrier layer is highly demanded. The high-temperature thermal stability and diffusion resistance of HEA nitrides makes it possible to be used as a diffusion barrier coating in conditions of elevated temperature and lower thickness. Jiang et al. [201] used TaTiAlCrZr/TaTiAlCrZrN film as diffusion barriers, and no intermetallic compound formed in the interconnect structure when annealed at 900 °C. A schematic of the structure is shown in Figure 21. A single layer of (TaTiAlCrZr)$N_x$ HEA–nitride film with a total thickness of only 4.0 nm demonstrated a good diffusion barrier ability [201]. The HEA–nitride layers' high diffusion resistance, as well as the thermal stability, are mainly credited to the amorphous structure, multi-element effect, and high stacking density with no rapid

diffusion path. It was proved that the HEA–nitride layer possesses high diffusion resistance and can be used as diffusion barrier materials of next-generation Cu interconnection.

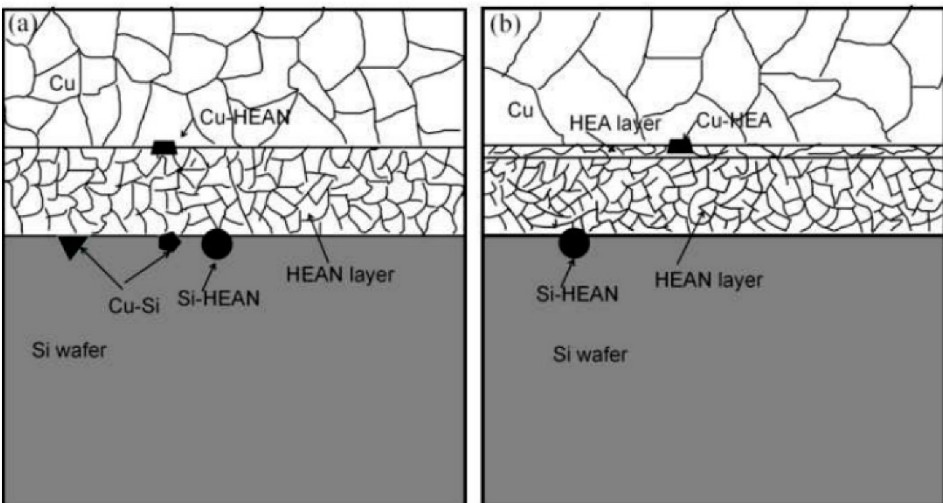

**Figure 21.** Shows an illustration of the deposited coating for controlling interdiffusion (**a**) shows diffusion pathways for $(AlCrTaTiZr)N_x$ (**b**) diffusion pathways of $(AlCrTaTiZr)/(AlCrTaTiZr)N_x$ [201].

Sluggish kinetic diffusion is one of the four core effects in HEMs and suppressed kinetic diffusion in HEAs is explained from both structural and thermodynamic perspectives. However, the compositional complexity of these materials makes diffusion measurement difficult, and conclusions are drawn from secondary observations (i.e., structure based) such as the appearance of nanocrystals, which is interpreted to signify slow diffusion. Thermodynamically, diffusion constants and activation energies have never been probed directly. The reason is the variety of atoms surrounding each lattice site, since vacancy formation and migration enthalpies are dependent on local atomic interaction. Studies on the fluctuation of lattice potential energies are limited to binary systems, and its investigation for higher-order systems is a challenge.

In summary, coatings based on HEMs materials have shown great potential as functional coatings having properties of corrosion, oxidation, radiation, and diffusion resistance. The distinguished functional properties of HEM coatings are primarily resulting from the core effects. The composition regulation and manufacturing processes could lead to a potentially wider range of properties of HEM coatings, broadening the functional applications in various fields where conventional materials have been applied.

## 5. Suggested Future Work

It is tempting to focus the investigation of the high-temperature phase stability of high-entropy materials (HEMs) on their application as high-temperature functional coating materials. Therefore, to grasp alloying rules, processing, and aggressive environmental effects and to gain an in-depth fundamental understanding of HEMs, and to realize their usage for functional applications, future investigations are suggested below:

(1) According to the design principles, such as consisting of single solid solutions, the inclusion of the anti-corrosion elements, and strong bonding, homogeneous, and densified microstructures, in-depth research should be carried out on the design of high-entropy coatings (HECs) with superior corrosion resistance.

(2) To meet the demands at high temperatures, it is necessary to investigate the oxidation properties of HECs by altering alloying additions and tailoring their microstructure to comprehend and establish fundamental theories/mechanisms of HEM coatings involved in its oxidation behaviour.

(3) To understand the radiation control mechanism at elevated temperature and high doses of irradiation, a set of different HECs needs to be investigated to correlate HEM intrinsic properties with defect dynamics of radiation-tolerant material.

(4) There is limited literature on the modelling and simulations of HEM films and coatings, which help explain the complex relationships among the preparation methods, microstructures, and properties. Further studies associated with the predictive computational modelling of the HEM films and coatings are urgently required.

## 6. Conclusions

This article reviewed research progress on high-entropy materials as a coating material for high-temperature applications based on outstanding physical, mechanical, and functional properties. The classification of various high-entropy coatings (HECs) discussed and highlighted the important elements that give each group unique properties. The main contents of this paper are summarised as follows:

(1) It is concluded that relative to HEM bulk preparation technologies, the required mechanical and functional properties can be easily achieved in HECs owing to their smaller thickness, hence the more rapid cooling rate.

(2) Similar to HEM bulk materials, the HECs are also have the tendency to form the solid-solution phase or amorphous phase due to the high-entropy effect and the 'fast quenching' of coating processes. The formation of the single solid-solution phase is discussed regarding the four core effects.

(3) The functional properties of HECs are that they exhibit excellent corrosion resistance, oxidation resistance, diffusion retardation, and high phase stability at elevated temperature.

(4) Several critical issues related to the reasons and design criteria for achieving the excellent functional and mechanical properties of the HECs are suggested, including the effects of stable oxide-forming elements on oxidation resistance as well as strong and non-nitride-forming elements on the hardness of HENs (high entropy nitrides).

**Author Contributions:** Conceptualization, M.B. and M.A. (Muhammad Arshad); writing—original draft preparation, M.A. (Muhammad Arshad); writing—review and editing, M.A. (Muhammad Arshad), M.A. (Mohamed Amer), Q.H., V.J., X.Z., M.M.; supervision, M.B. All authors have read and agreed to the published version of the manuscript.

**Funding:** This research received no external funding.

**Institutional Review Board Statement:** Not applicable.

**Informed Consent Statement:** Not applicable.

**Data Availability Statement:** This is a review article, and the entire data are presented within the article. The data presented in the figures are available on request from the corresponding author.

**Conflicts of Interest:** The authors declare no conflict of interest.

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
