# Peer review of "High-Entropy Coatings (HEC) for High-Temperature Applications: Materials, Processing, and Properties"

_coatings, doi:10.3390/coatings12050691_

Round 1
Reviewer 1 Report
Review report on the topic ‘High Entropy Coatings (HEC) for High Temperature Applications: Materials, Processing, and Properties’. Comments are listed below:
- Strengthen the abstract section. Add the key conclusion of the works in the last two lines of the abstract section.
- Discuss the novelty of the work in respect of the application.
- There are numerous spelling and grammatical errors. Please revise the manuscript thoroughly. Sentences are also not complete and references are also cited in a rough manner.
- Try to make a bridge between current and previously published work and specify the gap area and objective of the work. Refer to some recently published work on coating: https://doi.org/10.1007/s12540-020-00705-w; https://doi.org/10.1016/j.ceramint.2018.01.131; https://doi.org/10.1007/s41779-018-0258-4; https://orcid.org/0000-0002-3687-5226; 10.1520/JTE20180247.
- This is a review paper so a depth discussion and relevant references are required.
- Section related to corrosion and oxidation is very poorly discussed.
- Diffusion mechanisms need depth discussion and add more relevant figures.
- Shorten the length of the conclusion section. Keep only key points.
The work is good, but the technical discussion and introduction section needs improvement. Paper can be accepted after following minor corrections.
Reviewer 2 Report
As a review paper, this manuscript referenced almost entirely the literature review on the HEC for high-temperature applications. However, I cannot see a deep consideration or a detailed review of the literature. The authors just cited the keywords to the related references and there is no in-depth review. For a review paper, the literature shall be surveyed. What others exactly did has to be reviewed something like consideration of references 1 to 7. Nevertheless, the texts have been written in good English and no error has been observed. Graphical illustrations are of suitable quality. Generally, the paper is well-organized. My recommendation is a major revision based on giving more details of the literature (a true review of the literature).
Author Response
Dear Reviewer,
The authors greatly appreciate the effort and time that the reviewer has devoted to providing us with such valuable comments/suggestions on our manuscript. Indeed, the main driving force for conducting this review work was to highlight the current state of art in the field of high entropy coatings for high temperature applications and propose prospects for future research studies within this discipline. We agree with the high requirements of the review paper therefore we have done our best and made a comprehensive revision of the manuscript according to the reviewer’s suggestion in the past three weeks (initially we were given 10 days to revise). Some extra paragraphs were added in different sections and changes were made in existing sections of the manuscript and highlighted in red colour.
Reviewer 3 Report
The review paper ”High Entropy Coatings (HEC) for High Temperature Applications: Materials, Processing, and Properties” has the correct form for publication in Coatings Journal. The authors present the HEC Materials Classification and different types of HEA coatings with numerous references. The figures are very clear and the graphs are well presented. The authors synthesize all the information into 4 specific Conclusions.
Author Response
Dear Reviewer,
The authors greatly appreciate the effort and time that the reviewer has devoted to our manuscript. We highly appreciate the reviewer appreciation and encouragement of our work.
Round 2
Reviewer 2 Report
The revision is satisfactorily and my recommendation is to accept the paper.
Author Response
We highly appreciate the reviewer's appreciation and encouragement of our work.